# UQGAN: A Unified Model for Uncertainty Quantification of Deep Classifiers trained via Conditional GANs

**Philipp Oberdiek**
Department of Computer Science
TU Dortmund University
`philipp.oberdiek@cs.tu-dortmund.de`

**Gernot A. Fink**
Department of Computer Science
TU Dortmund University
`gernot.fink@cs.tu-dortmund.de`

**Matthias Rottmann**
School of Computer and Communication Sciences, EPFL
School of Mathematics and Natural Sciences, University of Wuppertal
`matthias.rottmann@epfl.ch`

## Abstract

We present an approach to quantifying both aleatoric and epistemic uncertainty for deep neural networks in image classification, based on generative adversarial networks (GANs). While most works in the literature that use GANs to generate out-of-distribution (OoD) examples only focus on the evaluation of OoD detection, we present a GAN based approach to learn a classifier that produces proper uncertainties for OoD examples as well as for false positives (FPs). Instead of shielding the entire in-distribution data with GAN generated OoD examples which is state-of-the-art, we shield each class separately with out-of-class examples generated by a conditional GAN and complement this with a one-vs-all image classifier. In our experiments, in particular on CIFAR10, CIFAR100 and Tiny ImageNet, we improve over the OoD detection and FP detection performance of state-of-the-art GAN-training based classifiers. Furthermore, we also find that the generated GAN examples do not significantly affect the calibration error of our classifier and result in a significant gain in model accuracy.

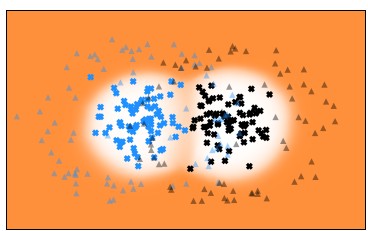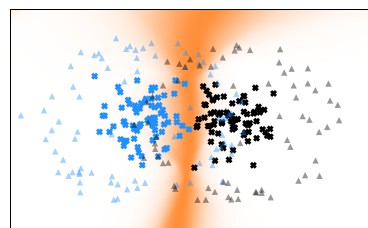

Figure 1: Toy example of two slightly overlapping Gaussian distributions. From left to right: 1. OoD heatmap with orange indicating a high probability of being OoD and white of in-distribution; 2. Aleatoric uncertainty (entropy over Equation (3)) with orange indicating high and white low uncertainty. Triangles indicate GAN-generated out-of-class examples and crosses correspond to the in-distribution data, while their color is coding the class membership.

36th Conference on Neural Information Processing Systems (NeurIPS 2022).

# 1 Introduction

Deep learning has shown outstanding performance in image classification tasks [22, 43, 13]. However, due to the enormous capacity and the inability of rejecting examples, deep neural networks (DNNs) are not capable of expressing their uncertainties appropriately. DNNs have demonstrated the tendency to overfit training data [46] and to be easily fooled into wrong class predictions with high confidence [9, 12]. Also, importantly, ReLU networks have been proven to show high confidence far away from training data [14]. This is contrary to the behavior that would be expected naturally by humans, which is to be uncertain when being confronted with new data examples that have not been observed during training.

Several approaches to predictive uncertainty quantification have been introduced in recent years, considering uncertainty in a Bayesian sense [2, 7, 18] as well as from a frequentist's point of view [15, 5, 34]. A common evaluation protocol is to discriminate between true and false positives (FPs) by means of a given uncertainty quantification. For an introduction to uncertainty in machine learning, we refer to [17], for a survey on uncertainty quantification methods for DNNs see [8].

By design of ordinary DNNs for image classification, their uncertainty is often studied on in-distribution examples [1]. The task of out-of-distribution (OoD) detection (or novelty detection) is oftentimes considered separately from uncertainty quantification [29, 44, 31]. Thus, OoD detection in deep learning has spawned an own line of research and method development. Among others, changes in architecture [5], loss function [41, 48] and the incorporation of data serving as OoD proxy [16] have been considered in the literature. Generative adversarial networks (GANs) have been used to replace that proxy by artificially generated data examples. In [27], examples of OoD data are created such that they shield the in-distribution regime from the OoD regime. Note that this often requires pre-training of the generator. E.g. in the aforementioned work, the authors constructed OoD data in their 2D example to pretrain the generator. While showing promising results, such GAN-based approaches mostly predict a single score for OoD detection and do not yield a principled approach to uncertainty quantification distinguishing in-distribution uncertainty between classes and out-of-distribution uncertainty.

In this work, we propose to use GANs to, instead of shielding all classes at once, shield each class separately from the out-of-class (OoC) regime (also cf. figure 1). Instead of maximizing an uncertainty measure, like softmax entropy, we combine this with a one-vs-all classifier in the final DNN layer. This is learned jointly with a class-conditional generator for out-of-class data in an adversarial framework. The resulting classifiers are used to model (class conditional) likelihoods. Via Bayes rule we define posterior class probabilities in a principled way. Our work thus makes the following novel contributions:

1. We introduce a GAN-based model yielding a classifier with complete uncertainty quantification.

2. Our model allows to distinguish uncertainty between classes (in large sample limit, if properly learned, approaching aleatoric uncertainty) from OoD uncertainty (approaching epistemic uncertainty).

3. By a conditional GAN trained with a Wasserstein-based loss function, we achieve class shielding in low dimensions without any pre-training of the generator.

4. In higher dimensions, we use a class conditional autoencoder and train the GAN on the latent space. This is coherent with the conditional GAN, allows us to use less complex generators and critics and reduces the influence of adversarial directions.

5. We improve over the OoD detection and FP detection performance of state-of-the-art GAN-training based classifiers.

We present in-depth numerical experiments with our method on MNIST, CIFAR10, CIFAR100 and Tiny ImageNet, accompanied with various OoD datasets. We outperform other approaches, also GAN-based ones, in terms of OoD detection and FP detection performance on CIFAR10. Also on MNIST, CIFAR100 and Tiny ImageNet we achieve superior OoD detection performance. Noteworthily, on the more challenging CIFAR10, CIFAR100 and Tiny ImageNet datasets, we achieve significantly stronger model accuracy compared to other approaches based on the same network architecture.

## 2 Related Work

In this section we give an overview of common uncertainty quantification methods as well as publications related to the different parts of our method. In this context the task of uncertainty quantification is to assign scalar values to predictions, quantifying aleatoric (in-distribution) and epistemic (out-of-distribution) uncertainty.

**(Baselines)** The works by [52, 15, 5] can be considered as early baseline methods for the task of OoD detection. They are frequentist approaches relying on confidence scores gathered from model outputs. The problem can oftentimes be attributed to the usage of the softmax activation function, which leads to overconfident predictions, in particular far away from training data, or to the decoupling of the confidence score from the original classification model during test time. Our proposed method does not make use of the softmax activation function and can produce unified uncertainty estimates during test time without the requirement of auxiliary confidence scores.

**(Conformal methods)** Methods associated with the term conformal predictions [35, 3, 30] are predicting, for a given confidence value, a set of classes that is likely to contain the real class. As such, they can give a more intuitive explanation to which classes might be getting confused. In essence, the uncertainty of a prediction can be derived from the size of the predicted set, assigning higher prediction uncertainty to larger sets. They are however not able to quantify both, aleatoric and epistemic uncertainty, and assigning a scalar uncertainty value to predictions, as in our setting, might not be straight forward.

**(Data perturbation and auxiliary data)** Many methods use perturbed training examples [28, 37, 29] or auxiliary outlier datasets [16, 20]. [28] use them for their confidence score based on class conditional Gaussian distributions, while [29] and [37] are utilizing them to increase the separability between in- and out-of-distribution examples. [16] use a hand picked auxiliary outlier dataset during model training and [20] use it for selecting a suitable GAN discriminator during training which then serves for OoD detection. The common problem with using perturbed examples is the sensitivity to hyperparameter selection which might render the resulting examples uninformative. Additionally, auxiliary outlier datasets cannot always be considered readily available and pose the problem of covering only a small proportion of the real world. In contrast, our method is able to produce OoC examples that are very close to the in-distribution but still distinguishable from it, thus we do not require explicit data perturbation or any auxiliary outlier datasets.

**(Bayesian methods)** Bayesian approaches [7, 2, 24] provide a strong theoretical foundation for uncertainty quantification and OoD detection. [24] propose deep ensembles that approximate a distribution over models by averaging predictions of multiple independently trained models. [7] are utilizing Monte-Carlo (MC) sampling with dropout applied to each layer. In [2] a variational learning algorithm for approximating the intractable posterior distribution over network weights has been proposed. While the theoretical foundation is strong, these methods often require changing the architecture, restricting the model space and/or increased computational cost. While making use of Bayes-rule, we are staying in a frequentist setting and are not dependent on sampling or ensemble techniques. This reduces computational cost and enables our model to produce high quality aleatoric and epistemic uncertainty estimates with a single forward pass. Also, our proposed framework does not change the network architecture, except for the output layer activation function, and thus makes it compatible with previously published techniques.

**(One-vs-All methods)** One-vs-All methods in the context of OoD detection have been recently studied by [6, 34, 38]. In the work by [6] an ensemble of binary neural networks is trained to perform one-vs-all classification on the in-distribution data which are then weighted by a standard softmax classifier. [34] use a DNN with a single sigmoid binary output for every class and explore the possibility of training the one-vs-all network with a distance based loss function instead of the binary cross entropy. Domain adaptation is considered in [38], where they utilize a one-vs-all classifier for a first OoD detection step before classifying into known classes with a second model. Their training objective is also accompanied by a hard negative mining on the in-distribution data. All these methods use the maximum predicted probability as a score for OoD detection and/or classification and do not aggregate the other probabilities into a single score like the method proposed by us. They also do not distinguish into different kinds of uncertainties as in our work. Lastly their training objectives are only based on in-distribution data. Generated OoC data as used in the present work is not considered.

**(Generative methods)** More recently generative model based methods [39, 27, 45, 47, 51] have shown strong performance on the task of out-of-distribution detection by supplying classification models with synthesized out-of-distribution examples. [39] utilize the latent space of a GAN by gradient based reconstruction of an input example. In the work by [27], a GAN architecture with an additional classification model is built. The classification model is trained to output a uniform distribution over classes on GAN examples close to the in-distribution. This approach is further improved by [45] who show improvements on the task of out-of-distribution detection. The generalization to distant regions in the sample space and the quality of generated boundary examples is however questionable [51]. A similar approach using a normalizing flow and randomly sampled latent vectors is proposed by [10]. The high level idea of the architectures proposed in the previously mentioned works is similar to the one proposed by us. However, other works are not able to approximate the boundary of data distributions with multiple modes as shown by [51]. Due to the fact that our GAN is class conditional and trained on a low dimensional latent space, we are able to follow multiple distribution modes resulting from different classes. We improve the in-distribution shielding by using a low-dimensional regularizer and have an additional advantage in terms of computational cost as our cGAN model architecture can be chosen with considerably smaller size due to it being trained in the latent space. Furthermore, these methods do not yield separate uncertainty scores for FP and OoD examples.

Generating OoD data based on lower dimensional latent representations has been explored in [50, 40]. [50] utilize a variational Autoencoder (vAE) to produce examples that are inside the encoded manifold (type I) as well as outside of it (type II). [40] (GEN) also use a vAE and train a GAN in the resulting latent space, assigning generated examples to the OoD domain in order to estimate a Dirichlet distribution on the class predictions. Utilizing a vAE has the advantage that one can make assumptions on the distribution of the latent space but also results in slightly blurry reconstructions. While the work of [40] is the most similar to our method, we improve on several shortcomings. First of all we employ class conditional models to improve diversity and class shielding. Additionally, we are able to distinguish aleatoric and epistemic uncertainty while the method by [40] is not. It assigns the same type of uncertainty to OoD and FP examples.

## 3 Method

### 3.1 One-vs-All Classification

We start by formulating our classification model as an ensemble of one-vs-all classifiers. Let $C(o|x, y)$ model the probability that for a given class $y \in \mathcal{Y} = \{1, \dots, n\}$, an example with features $x \in \mathcal{X} = \mathbb{R}^d$ is OoC. Analogously, $C(i|x, y) = 1 - C(o|x, y)$ for a given class $y$ models the probability of $x$ being in-class. Let $S \subseteq \mathcal{X} \times \mathcal{Y}$ be our training dataset with $\hat{p}(y)$, $y \in \mathcal{Y}$, the estimated relative class frequencies. To model $C(i|x, y)$, we use a DNN with $n \geq 2$ output neurons equipped with sigmoid activations. For each class output $y$, the data corresponding to class $y$ serves as in-class data and all other data as OoC data. Hence, a basic variant of our training objective is given by a weighted empirical binary cross entropy

$$\min_{C} \frac{1}{|S|} \sum_{(x,y) \in S} \left[ -\log(C(i|x,y)) - \frac{1}{n-1} \sum_{y' \in \mathcal{Y} \setminus \{y\}} \frac{\hat{p}(y)}{\hat{p}(y')} \log(C(o|x,y')) \right] . \tag{1}$$

Therein, $\frac{\hat{p}(y)}{\hat{p}(y')}$ is weighting the OoC loss to counter potential class imbalance. Similarly, $\frac{1}{n-1}$ weighs the OoC loss compared to the in-class loss. Applying the transformation

$$\tilde{C}(i|x,y) = \frac{\frac{1}{n} C(i|x,y)}{\frac{1}{n} C(i|x,y) + \frac{n-1}{n} C(o|x,y)} \tag{2}$$

after training, we obtain an appropriate classifier, formalized as follows:

**Lemma 1** (Class posterior). *Under typical assumptions of statistical learning theory and training $C$ on equation* (1) *it holds that for $|S| \to \infty$*

$$\hat{p}(y|x) = \frac{\tilde{C}(i|x,y)\hat{p}(y)}{\sum_{y' \in \mathcal{Y}} \tilde{C}(i|x,y')\hat{p}(y')} \longrightarrow p(y|x) . \tag{3}$$

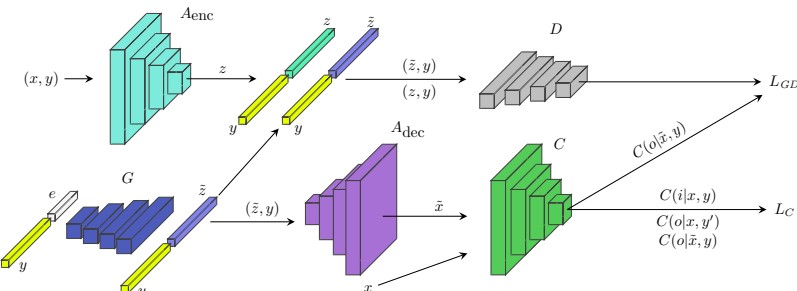

Figure 2: Overview of the proposed architecture. Before training the GAN objective together with the classifier, the cAE is pretrained on the in-distribution training dataset. After that the weights of $A_{\text{enc}}$ and $A_{\text{dec}}$ are frozen. The flow of information is from left to right, with the top attributed to the GAN loss $L_{GD}$ and the bottom to the classifier loss $L_C$.

See appendix A for the detailed assumptions and the proof, explaining our choice of the classification model. We distinguish between aleatoric and epistemic uncertainty. While aleatoric uncertainty is considered to be inherent to all observations and irreducible (for a fixed $\mathcal{X}$), epistemic uncertainty is induced by the model choice and the amount of data and can be reduced by additional data and appropriate model selection. Using $\hat{p}(y|x)$, we estimate the probability of an example $x$ being in-distribution by defining

$$\tilde{C}(i|x) = \sum_{y \in \mathcal{Y}} \tilde{C}(i|x,y)\hat{p}(y|x) = \sum_{y \in \mathcal{Y}} \frac{\tilde{C}(i|x,y)^2 \hat{p}(y)}{\sum_{y' \in \mathcal{Y}} \tilde{C}(i|x,y')\hat{p}(y')}, \quad (4)$$

which yields a quantification of epistemic uncertainty via $\tilde{C}(o|x) = 1 - \tilde{C}(i|x)$. For aleatoric uncertainty estimation, we consider the Shannon entropy of the predicted class probabilities

$$H(x) = -\sum_{y \in \mathcal{Y}} \hat{p}(y|x) \log(\hat{p}(y|x)). \quad (5)$$

This is a sensible definition since we showed that $\hat{p}(y|x) \to p(y|x)$ for $|S| \to \infty$. If the real $p(y|x)$ is close to a uniform distribution over the classes (which maximizes the Shannon entropy) we have a high uncertainty and vice versa. For epistemic uncertainty $\tilde{C}(i|x)$ can be considered as a proxy for $p(i|x)$, which results in the epistemic uncertainty by $1 - \tilde{C}(i|x)$. In addition, we generate for each class OoC examples with a conditional GAN. A joint training of classifier and GAN is introduced in the upcoming section 3.2.

## 3.2 GAN Architecture

Similar to [27], we combine our classification model with a GAN and train both alternatingly. The assumptions that we made for lemma 1 are standard in statistical learning theory, however it is important to notice that we exclude the GAN-generated data from the statement. By incorporating GAN-generated data into the training of our classification model introduced in section 3.1, we are violating this assumption. However, in section 4 we demonstrate empirically that this does not harm our classification model performance-wise. The Wasserstein GAN with gradient penalty proposed by [11] serves a basis for our conditional GAN (cGAN). Additionally, we condition the generator as well as the critic on the class labels to generate class specific OoC examples. Inspired by [40, 50] where the latent space of a variational autoencoder (vAE) is utilized for GAN training, we proceed analogously, however using a vanilla conditional autoencoder (cAE) as we observed reconstructions with higher visual quality for this choice. Training the cGAN on a latent space, we do not generate adversarial noise examples and can use less complex generators and critics. Prior to the cGAN training, we train the cAE on in-distribution data and then freeze the weights during the cGAN and classifier training. The optimization objective of the cAE is given as pixel-wise binary cross-entropy

$$\min_A \frac{1}{|S|} \sum_{(x,y) \in S} -\frac{1}{N_x} \sum_{i=1}^{N_x} x_i \cdot \log(\hat{x}_i) + (1 - x_i) \cdot \log(1 - \hat{x}_i), \quad (6)$$

with $\hat{x}_i = A_{\text{dec}}(z, y)$ the decoded latent variable, $z = A_{\text{enc}}(x, y)$ being the encoded example, $x_i$ the $i$-th pixel of example $x$ and $N_x$ the number of pixels belonging to $x$. Therein, the pixel values are assumed to be in the interval $[0, 1]$, while $0 \cdot \log(0) = 0$. The cGAN is trained using the objective function

$$\min_G \max_D \quad \frac{1}{|S|} \sum_{(x,y) \in S} D(z|y) - D(\tilde{z}|y) + \lambda_{gp} \cdot L_{gp}, \tag{7}$$

with $D$ the conditional critic, $z = A_{\text{enc}}(x, y)$, $\tilde{z} = G(e, y)$ the latent embedding produced by the conditional generator, $e \sim U(0, 1)$ noise from a uniform distribution, $y$ a class label and $L_{gp}$ the gradient penalty from [11]. Integrating the classification objective into the cGAN objective, we alternate between

$$\min_G \max_D \overbrace{\frac{1}{|S|} \sum_{(x,y) \in S} D(z|y) - D(\tilde{z}|y) + \lambda_{gp} \cdot L_{gp} - \lambda_{cl} \cdot \log(C(o|\tilde{x}, y)) + \lambda_R L_R}^{L_{GD}} \tag{8}$$

and

$$\min_C \frac{1}{|S|} \sum_{(x,y) \in S} \overbrace{\left[ -\log(C(i|x,y)) - \frac{\lambda_{\text{real}}}{n-1} \left( \sum_{y' \in \mathcal{Y} \setminus \{y\}} \frac{\hat{p}(y)}{\hat{p}(y')} \log(C(o|x, y')) \right) \right.}^{L_C}$$
$$\left. - (1 - \lambda_{\text{real}}) \cdot \log(C(o|\tilde{x}, y)) \right], \tag{9}$$

with $\lambda_{\text{real}} \in [0, 1]$ being an interpolation factor between real and generated OoC examples and $L_R$ being an additional regularization loss for the generated latent codes with hyperparameter $\lambda_R \geq 0$, which we introduce in section 3.3. The latent embeddings produced by the cGAN are decoded with the pretrained cAE, thus $\tilde{x} = A_{\text{dec}}(\tilde{z}, y) = A_{\text{dec}}(G(e, y), y)$. That is, the cGAN is trained on the latent space while the classification model is trained on the original feature space. Our entire GAN-architecture is visualized in figure 2.

### 3.3  Low-Dimensional Regularizer

In low dimensional latent spaces we found it to be advantageous to apply an additional regularizer to the generated latent embeddings $\tilde{z}$ to improve class shielding. Let $(z, y)$ be the latent embedding of an example $x$ with its corresponding class label $y$ and $\mathcal{Z}(z, y) = \{\tilde{z} - z | \tilde{z} = G(e, y), e \sim U(0, 1)\} = \left\{ \bar{z}^1, \ldots, \bar{z}^{N_y^z} \right\}$ all generated latent codes with the same class label and normalized to origin $z$. We encourage the generator to produce latent codes that more uniformly shield the class $y$ by maximizing the average angular distance between all $\bar{z} \in \mathcal{Z}(z, y)$, which corresponds to minimizing

$$l_R(z, y) = \frac{2}{N_y^z \cdot (N_y^z - 1)} \cdot \sum_{\substack{\bar{z}^i, \bar{z}^j \in \mathcal{Z}(z, y) \\ i < j}} -\log \left( \arccos \left( \frac{\bar{z}^i * \bar{z}^j}{\|\bar{z}^i\| \cdot \|\bar{z}^j\|} \right) \cdot \frac{1}{\pi} \right), \tag{10}$$

with $*$ being the dot-product. The logarithm introduces an exponential scaling for very small angular distances, encouraging a more evenly spread distribution of the generated latent codes. This loss is then averaged over all class labels and training data examples

$$L_R = \frac{1}{n} \sum_{y \in \mathcal{Y}} \left[ \frac{1}{N_y} \sum_{(x,y) \in S} l_R(A_{\text{enc}}(x, y), y) \right], \tag{11}$$

with $N_y = |\{(x, y')|y' = y\}|$ the number of examples with class label $y$. In appendix B a more detailed explanation of this regularization loss is provided.

During our experiments we studied different regularizer losses such as Manhattan/Euclidean distance, infinity norm and standard cosine similarity. In experiments, we found that for our purpose equation (11) performed best. We argue that this can be attributed to the independence of the latent space

| Method | Reference |
|---|---|
| Maximum softmax | [15] |
| Entropy | |
| Bayes-by-Backprop | [2] |
| MC-Dropout | [7] |
| Deep-Ensembles | [24] |
| Confident Classifier | [27] |
| GEN | [40] |

Table 1: Baselines and related methods used for comparing our work.

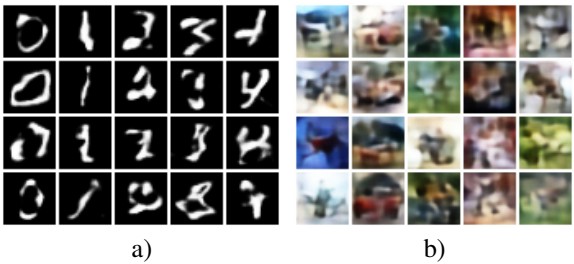

a)  b)

Figure 3: Generated OoC examples by our approach. a) MNIST examples for digit classes 0-4 (left to right). b) CIFAR10 examples for classes *airplane*, *automobile*, *bird*, *cat* and *deer* (left to right). See figure 5 for more examples.

value range, which can have a large impact on the $p$-norm distance metrics, and to the exponential scaling for very small angular distances.

To this end, the basic variant of our method does not account for model uncertainty. Thus, we also include results with MC-Dropout, which also demonstrates the compatibility of our model with existing methods. Another limitation of our approach is that we are not explicitly accounting for adversarial attacks because during training in the cAE latent space we are removing any adversarial directions.

## 4  Experiments

We compare our method with a number of related approaches (cf. table 1). While the first two methods are simple baselines, the subsequent three ones are Bayesian ones and the final two methods are GAN-based (cf. section 2). Following the publications [40, 50, 37, 15], we consider four experimental setups using the MNIST ($28 \times 28$, 10 classes) [26], CIFAR10 ($32 \times 32$, 10 classes) [21], CIFAR100 ($32 \times 32$, 100 classes) [21] and Tiny ImageNet ($64 \times 64$, 200 classes) [25] datasets as in-distribution, respectively. Similar to [40, 50] and others, we split the datasets class-wise into two non-overlapping sets, i.e., MNIST 0-4 / 5-9, CIFAR10 0-4 / 5-9, CIFAR100 0-49 / 50-99 and Tiny ImageNet 0-99 / 100-199. While the first half serves as in-distribution data, the second half constitutes OoD cases close to the training data and therefore difficult to detect. For the MNIST 0-4 dataset, we consider the MNIST 5-9, EMNIST-Letters [4], Fashion-MNIST [53], Omniglot [23], SVHN [33] and CIFAR10 datasets as OoD examples. For the CIFAR10 0-4 datasets, we use CIFAR10 5-9, LSUN [55], SVHN, Fashion-MNIST and MNIST as OoD examples. The same OoD datasets are used for CIFAR100 except CIFAR10 5-9 being replaced by CIFAR100 50-99. For the Tiny ImageNet 0-99 dataset we use Tiny ImageNet 100-199, SVHN, Fashion-MNIST and MNIST as OoD examples. These selections yield compositions of training and OoD examples with strongly varying difficulty for state-of-the-art OoD detection. Besides that, we examine our method's behavior on a 2D toy example with two overlapping Gaussians (having trivial covariance structure), see figure 1. Additionally, we split the official training sets into $80\%$ / $20\%$ training / validation sets, where the latter are used for hyperparameter tuning and model selection. Like related works, we utilize the LeNet-5 architecture on MNIST and a small ResNet on CIFAR10, CIFAR100 and Tiny ImageNet as classification models. To ensure fair conditions, we re-implemented all aforementioned methods while following the authors recommendations for hyperparameters and their reference implementations. For methods involving more complex architectures, e.g. a GAN or a VAE as in [27, 40], we used the proposed architectures for those components, while for the sake of comparability sticking to our choice of classifier models. All implementations are using PyTorch [36] and can be found in the associated repository[1]. For each method, we selected the network checkpoint with maximal validation accuracy during training. For a more detailed overview of the hyperparameters used in our experiments, we refer to appendix D or the implementation.

For evaluation we use the following well established metrics:

- Classification accuracy on the in-distribution datasets.

---

[1] https://github.com/RonMcKay/UQGAN

Table 2: Results for MNIST (0-4) as in-distribution vs {MNIST (5-9), EMNIST-Letters, Omniglot, Fashion-MNIST, SVHN, CIFAR10} as out-of-distribution datasets

| Method | In-Distribution | | | Out-of-Distribution | | | |
|---|---|---|---|---|---|---|---|
| | Accuracy ↑ | AUROC S/F ↑ | ECE ↓ | AUROC ↑ | AUPR-In ↑ | AUPR-Out ↑ | FPR@95% TPR ↓ |
| Ours | 99.74 (0.05) | 99.35 (0.31) | 0.15 (0.05) | 98.03 (0.28) | 80.05 (2.65) | 99.87 (0.02) | 8.73 (1.47) |
| Ours with MC-Dropout | 99.80 (0.04) | 99.42 (0.11) | 1.38 (0.11) | **98.58 (0.25)** | **83.71 (2.40)** | **99.91 (0.02)** | **5.60 (0.77)** |
| One-vs-All Baseline | 99.84 (0.06) | **99.84 (0.06)** | 0.12 (0.04) | 97.12 (0.17) | 66.68 (1.93) | 99.81 (0.01) | 9.45 (0.56) |
| Max. Softmax [15] | 99.87 (0.02) | 99.68 (0.13) | **0.11 (0.02)** | 97.07 (0.12) | 69.00 (1.65) | 99.81 (0.01) | 9.71 (0.37) |
| Entropy | 99.87 (0.02) | 99.66 (0.14) | **0.11 (0.02)** | 97.13 (0.12) | 68.76 (1.94) | 99.81 (0.01) | 9.65 (0.39) |
| Bayes-by-Backprop [2] | 99.67 (0.02) | 99.50 (0.06) | 0.78 (0.05) | 95.46 (0.26) | 67.09 (2.70) | 99.60 (0.03) | 17.33 (1.06) |
| MC-Dropout [7] | 99.91 (0.02) | 99.62 (0.12) | 0.83 (0.10) | 97.69 (0.16) | 72.82 (2.55) | 99.86 (0.01) | 8.28 (0.39) |
| Deep-Ensembles [24] | **99.89 (0.03)** | 99.74 (0.08) | 0.15 (0.01) | 97.70 (0.03) | 73.09 (0.62) | 99.86 (0.00) | 7.81 (0.21) |
| Confident Classifier [27] | 99.82 (0.02) | 99.62 (0.15) | 0.16 (0.01) | 98.15 (0.13) | 78.31 (2.01) | 99.88 (0.01) | 7.65 (0.46) |
| GEN [40] | 99.70 (0.03) | 98.13 (0.45) | 1.98 (0.85) | 97.78 (0.70) | 69.77 (10.20) | 99.86 (0.04) | 8.98 (1.90) |
| Entropy Oracle | 99.79 (0.06) | 98.93 (0.68) | 0.82 (0.06) | 99.90 (0.02) | 98.66 (0.28) | 99.99 (0.00) | 0.43 (0.10) |
| One-vs-All Oracle | 99.77 (0.03) | 99.47 (0.16) | 0.14 (0.02) | 99.90 (0.01) | 98.50 (0.12) | 99.99 (0.00) | 0.40 (0.03) |

Table 3: Results for CIFAR10 (0-4) as in-distribution vs {CIFAR10 (5-9), LSUN, SVHN, Fashion-MNIST, MNIST} as out-of-distribution datasets

| Method | In-Distribution | | | Out-of-Distribution | | | |
|---|---|---|---|---|---|---|---|
| | Accuracy ↑ | AUROC S/F ↑ | ECE ↓ | AUROC ↑ | AUPR-In ↑ | AUPR-Out ↑ | FPR@95% TPR ↓ |
| Ours | 87.26 (0.29) | 84.71 (0.52) | 10.35 (0.20) | 86.49 (0.63) | 49.08 (1.05) | 98.72 (0.08) | 45.78 (2.90) |
| Ours with MC-Dropout | **90.26 (0.22)** | **89.19 (0.13)** | **2.34 (0.33)** | **89.64 (0.23)** | **53.15 (0.27)** | **99.01 (0.03)** | **43.54 (1.56)** |
| One-vs-All Baseline | 82.82 (0.62) | 82.19 (0.81) | 8.62 (4.55) | 72.52 (2.16) | 32.24 (2.14) | 96.01 (0.42) | 88.74 (1.86) |
| Max. Softmax [15] | 82.42 (0.31) | 83.29 (0.89) | 11.34 (0.83) | 72.52 (0.51) | 30.52 (0.97) | 96.10 (0.05) | 87.68 (0.43) |
| Entropy | 82.42 (0.31) | 83.41 (0.88) | 11.34 (0.83) | 72.85 (0.49) | 30.43 (0.87) | 96.21 (0.06) | 85.41 (0.89) |
| Bayes-by-Backprop [2] | 84.05 (0.33) | 85.22 (0.40) | 9.17 (0.41) | 74.23 (0.96) | 29.91 (1.61) | 96.48 (0.19) | 83.97 (1.47) |
| MC-Dropout [7] | 85.08 (0.56) | 83.91 (0.49) | 9.90 (0.42) | 77.56 (1.27) | 38.75 (1.40) | 96.85 (0.17) | 82.35 (1.08) |
| Deep-Ensembles [24] | 85.43 (0.22) | 85.29 (0.57) | 3.10 (0.29) | 74.24 (0.73) | 32.81 (1.50) | 96.43 (0.10) | 85.07 (0.81) |
| Confident Classifier [27] | 83.58 (0.11) | 85.08 (0.18) | 9.31 (0.80) | 73.33 (0.53) | 32.32 (1.09) | 96.29 (0.11) | 85.04 (1.09) |
| GEN [40] | 82.46 (0.35) | 82.88 (0.49) | 6.71 (1.81) | 86.01 (1.60) | 42.32 (2.81) | 98.66 (0.17) | 45.39 (3.26) |
| Entropy Oracle | 83.41 (0.58) | 80.73 (0.54) | 7.53 (0.27) | 95.44 (0.28) | 68.51 (0.57) | 99.57 (0.04) | 17.27 (1.44) |
| One-vs-All Oracle | 83.70 (0.50) | 81.27 (1.03) | 8.85 (0.49) | 91.38 (0.74) | 53.75 (1.49) | 99.15 (0.08) | 35.94 (3.79) |

- Area under the Receiver Operating Characteristic Curve (AUROC). We apply the AUROC to the binary classification tasks in-/out-of-distribution (via the score from equation (4)) and TP/FP (Success/Failure) (via the score from equation (5)).
- Expected Calibration Error (ECE) [32] applied to the estimated class probabilities $\hat{p}(y|x)$ for in-distribution examples $x$, computed on 15 bins.
- Area under the Precision Recall Curve (AUPR) w.r.t. the binary in-/out-of-distribution decision (via the score from equation (4)). We further distinguish between AUPR-In and AUPR-Out. For AUPR-in the in-distribution class is the positive one, while for AUPR-Out the out-of-distribution class is the positive one.
- FPR @ 95% TPR computes the False Positive Rate (FPR) at the decision threshold on the OoD score from equation (4) ensuring a True Positive Rate (TPR) of 95%.

Before discussing our results on MNIST, CIFAR10, CIFAR100 and Tiny ImageNet, we briefly discuss our findings on the 2D example. As can be seen in figure 1, the generated OoC examples are nicely shielding the respective in-distribution classes. OoC examples of one class can be in-distribution examples of other classes. This is an intended feature and to this end, the loss term for the synthesized OoC examples in equation (9) is class conditional. This feature is supposed to make our one-vs-all classifier predict a high OoC probability in the OoC regime. One can also observe that the estimated epistemic and aleatoric uncertainties are complementary, resulting in a high aleatoric uncertainty in the overlapping region of the Gaussians while also having a low epistemic uncertainty there. This is one of the main advantages that sets our approach apart from related methods. Results for a slightly more challenging 2D toy example on the two moons dataset and on a grid of Gaussians are presented in appendix C. We now demonstrate that this result generalizes to higher dimensional problems.

For MNIST and CIFAR10, figure 3 shows the OoC examples produced at the end of the generator training. Due to using a conditional GAN and an AE, we are able to generate OoC examples (instead of only out-of-distribution as in related works) during test time. It can be seen that the resulting examples resemble a lot of semantic similarities with the original class while still being distinguishable from them.

Table 4: Results for CIFAR100 (0-49) as in-distribution vs {CIFAR100 (50-99), LSUN, SVHN, Fashion-MNIST, MNIST} as out-of-distribution datasets

| Method | In-Distribution | | | Out-of-Distribution | | | |
|---|---|---|---|---|---|---|---|
| | Accuracy ↑ | AUROC S/F ↑ | ECE ↓ | AUROC ↑ | AUPR-In ↑ | AUPR-Out ↑ | FPR@95% TPR ↓ |
| Ours | 56.60 (0.73) | 73.61 (0.33) | 34.32 (0.42) | 80.11 (1.40) | 28.81 (1.32) | 97.98 (0.19) | **55.23 (3.20)** |
| Ours with MC-Dropout | **64.53 (0.47)** | 80.38 (0.40) | 10.92 (0.24) | **80.75 (1.19)** | **31.75 (1.53)** | 98.04 (0.14) | 58.10 (2.11) |
| One-vs-All Baseline | 50.90 (0.83) | 76.21 (0.91) | 23.89 (2.16) | 62.99 (1.59) | 15.69 (2.69) | 94.53 (0.25) | 92.44 (0.76) |
| Max. Softmax [15] | 56.12 (0.60) | 80.68 (0.29) | 19.10 (3.23) | 67.68 (1.56) | 23.03 (1.71) | 95.47 (0.30) | 88.42 (1.42) |
| Entropy | 56.12 (0.60) | 81.16 (0.26) | 19.10 (3.23) | 69.43 (1.71) | 23.75 (1.85) | 95.77 (0.32) | 87.08 (1.74) |
| Bayes-by-Backprop [2] | 56.02 (0.50) | 81.90 (0.55) | 14.71 (0.31) | 69.74 (0.76) | 24.60 (1.25) | 95.86 (0.12) | 87.01 (0.75) |
| MC-Dropout [7] | 59.88 (0.81) | 82.57 (0.60) | 21.94 (0.58) | 67.75 (1.15) | 22.31 (1.14) | 95.40 (0.22) | 89.38 (1.30) |
| Deep-Ensembles [24] | 62.36 (0.43) | **82.77 (0.47)** | **2.72 (0.37)** | 74.29 (0.50) | 29.38 (0.67) | 96.53 (0.13) | 83.37 (1.41) |
| Confident Classifier [27] | 54.16 (0.13) | 81.07 (0.44) | 27.19 (5.45) | 68.66 (0.48) | 22.51 (0.20) | 95.75 (0.08) | 86.17 (0.55) |
| GEN [40] | 51.16 (0.26) | 77.83 (0.80) | 41.26 (2.05) | 77.43 (2.58) | 26.12 (2.41) | 97.64 (0.33) | 61.48 (4.33) |
| Entropy Oracle | 53.66 (0.39) | 81.39 (0.69) | 19.82 (0.34) | 86.16 (0.41) | 38.63 (0.58) | 98.68 (0.05) | 44.67 (1.46) |
| One-vs-All Oracle | 51.82 (1.81) | 76.43 (0.83) | 15.26 (4.27) | 92.91 (0.28) | 47.68 (1.37) | 99.39 (0.03) | 22.35 (0.71) |

Table 5: Results for Tiny ImageNet (0-99) as in-distribution vs {Tiny ImageNet (100-199), SVHN, Fashion-MNIST, MNIST} as out-of-distribution datasets.

| Method | In-Distribution | | | Out-of-Distribution | | | |
|---|---|---|---|---|---|---|---|
| | Accuracy ↑ | AUROC S/F ↑ | ECE ↓ | AUROC ↑ | AUPR-In ↑ | AUPR-Out ↑ | FPR@95% TPR ↓ |
| Ours | 34.28 (0.37) | 71.90 (0.47) | 48.94 (0.66) | 79.25 (1.61) | 26.35 (2.25) | 97.66 (0.20) | 47.22 (1.85) |
| Ours with MC-Dropout | **45.60 (0.43)** | 79.18 (0.42) | 5.92 (0.38) | **94.96 (0.13)** | **59.76 (0.64)** | **99.51 (0.01)** | **13.72 (0.30)** |
| One-vs-All Baseline | 35.18 (0.26) | 76.23 (0.43) | 11.59 (4.41) | 55.19 (2.29) | 17.97 (2.41) | 90.97 (0.69) | 97.32 (0.62) |
| Max. Softmax [15] | 36.06 (0.30) | 78.56 (0.68) | 26.01 (7.25) | 61.53 (1.04) | 21.07 (0.98) | 93.29 (0.29) | 92.51 (0.87) |
| Entropy | 36.06 (0.30) | 79.39 (0.74) | 26.01 (7.25) | 62.44 (1.31) | 21.90 (0.86) | 93.16 (0.46) | 93.79 (1.28) |
| Bayes-by-Backprop [2] | 32.31 (0.43) | 78.44 (0.91) | 19.39 (0.62) | 68.05 (2.29) | 21.24 (2.14) | 95.23 (0.52) | 81.70 (3.53) |
| MC-Dropout [7] | 43.48 (0.53) | 80.63 (0.30) | **2.79 (0.35)** | 63.35 (4.23) | 27.11 (2.29) | 92.78 (1.06) | 95.86 (1.28) |
| Deep-Ensembles [24] | 42.48 (0.22) | **81.29 (0.38)** | 16.50 (6.94) | 67.76 (0.27) | 30.85 (0.42) | 93.93 (0.07) | 93.79 (0.23) |
| Confident Classifier [27] | 36.07 (0.53) | 78.66 (0.67) | 35.48 (2.13) | 59.99 (1.84) | 20.58 (1.61) | 92.66 (0.39) | 94.49 (0.51) |
| GEN [40] | 30.54 (0.84) | 73.40 (0.91) | 28.79 (0.79) | 84.65 (5.42) | 36.86 (8.25) | 98.28 (0.69) | 39.67 (12.50) |
| Entropy Oracle | 37.18 (0.50) | 79.50 (0.63) | 17.05 (2.82) | 85.43 (1.90) | 41.95 (2.31) | 98.27 (0.29) | 48.74 (6.63) |
| One-vs-All Oracle | 34.92 (0.56) | 75.10 (0.92) | 20.61 (3.50) | 95.75 (0.09) | 59.69 (0.50) | 99.59 (0.01) | 10.30 (0.46) |

All presented results are computed on the respective (official) test sets of the datasets. We also conducted an extensive parameter study on the validation sets, which is summarized in appendix E. The conclusion of this parameter study is that the performance of our framework is in general stable w.r.t. the choice of the hyperparameters. Increasing $\lambda_{reg}$ positively impacts the model performance up to a certain maximum. The best performance is obtained by choosing latent dimensions such that the cAE is able to compute reconstructions of good visual quality. Across all datasets, choosing $\lambda_{real} \in [0.5, 0.6]$ achieves the best detection scores, also indicating a positive influence of the generated OoC examples on the model's classification accuracy.

Tables 2 to 5 present the results of our experiments. The scores in the columns of the section *In-Distribution* are solely computed on the respective in-distribution dataset. The scores in the columns of the section *Out-of-Distribution* are displaying the OoD detection performance when presenting examples from the respective in-distribution dataset as well as from the entirety of all assigned OoD datasets. Note that we did not apply any balancing in the OoD datasets but included the respective test sets as is (see appendix D for the sizes of the test sets). As an upper bound on the OoD detection performance we also show results for two oracle models, supplied with the real OoD training datasets they are evaluated on. One of them is trained with the standard softmax and binary-cross-entropy to maximize entropy on OoD examples and the other one with our proposed loss function, cf. equation (9).

We first discuss the in-distribution performance of our method. W.r.t. MNIST, the results given in the left section of table 2 show that we are on par with state-of-the-art GAN-based approaches while still having a similar ECE, only being surpassed by Deep-Ensembles and the other baselines by a fairly small margin. However, considering the respective CIFAR10 results in the left section of table 3, we clearly outperform state-of-the-art GAN-based methods as well as all other baseline methods by a large margin. Noteworthily, we achieve an accuracy of $90.26\%$ which is 5 to 8 percent points (pp.) above the other classifiers and an AUROC S/F of $89.19\%$ which is 4 to 6 pp. higher than for the other methods. This corresponds to a relative improvement of $33\%$ in accuracy and $26\%$ in AUROC S/F compared to the second best method. A similar improvement in accuracy can be observed for CIFAR100 and Tiny ImageNet (tables 4 and 5) where the AUROC S/F is slightly lower but still on par with other approaches. Considering ECE on CIFAR10/100 and Tiny ImageNet, our MC-Dropout

variant is outperforming nearly all related methods, except for CIFAR100 and Tiny ImageNet where it is ranked second. This signals that, although we incorporate generated OoC examples impurifying the distribution of training data presented to the classifier, empirically there is no evidence that this harms the learned classifier, neither w.r.t. calibration, nor w.r.t. separation.

Considering the OoD results from the right-hand sections of tables 2 to 5, the superiority of our method compared to the other ones is now consistent over all four in-distribution datasets. On the MNIST dataset we are outperforming previously published works, especially considering the AUPR-In and FPR @ 95% TPR metrics, with a 25% and 27% relative improvement over the second best method, respectively. This is consistent with the results for CIFAR10 and CIFAR100 as in-distribution datasets where we achieve for both AUROC and AUPR-In a relative improvement of 19%–26% and 3%–15%, respectively, over the second best method. On the higher dimensional Tiny ImageNet dataset, which has $20\times$ more classes than MNIST/CIFAR10 and $4\times$ more input pixels, the results are even stronger with an 36%–67% relative improvement over the second best competitor.

Comparing our results with the ones of the oracles, two observations become apparent. Firstly, in some OoD experiments the GAN-generated OoC examples achieve results fairly close to or even surpassing the ones of the oracles while also in some of them there is still room for improvement left (in particular w.r.t. FPR@95%TPR). Secondly, GAN-generated OoC examples can help improve generalization (in terms of classification accuracy) while real OoD data might be too far away from the in-distribution data. An OoD-dataset-wise breakdown of the results in tables 2 to 5 is provided in appendix G. For MNIST, this breakdown reveals that our method performs particularly well in the difficult task of separating MNIST 0-4 and MNIST 5-9. On the other MNIST-related tasks we achieve mid-tier results, being slightly behind the other GAN-based methods. The same holds for CIFAR100 0-49 and CIFAR100 50-99. However, with regards to CIFAR10 and Tiny ImageNet we are consistently outperforming the other methods by large margins.

In appendix F we present results of an experiment where we perform FP and OoD detection jointly on the computed uncertainty scores. The main observations there are that our method outperforms other GAN-based methods and that our method including dropout achieves the overall best detection performance.

## 5  Conclusion

In this work, we introduced a GAN-based model yielding a one-vs-all classifier with complete uncertainty quantification. Our model distinguishes uncertainty between classes (in large sample limit approaching aleatoric uncertainty) from OoD uncertainty (approaching epistemic uncertainty). We have demonstrated in numerical experiments that our model sets a new state-of-the-art w.r.t. OoD as well as FP detection. The generated OoC examples do not harm the training success in terms of calibration, but even improve it in terms of accuracy. We have seen that incorporating MC dropout to account for model uncertainty can further improve the results.

**Broader Impact**

We are contributing generally applicable methodology which can be used in any real world application. The general direction of making the deployment of models more feasible due to improved quality of uncertainties might ultimately result in a reduction of the number of jobs in the respective sectors. However, as of now, it is also important to note that this method cannot give any guarantees on the estimated uncertainties and thus these should be used with caution and never solely relied upon.

**Acknowledgment**

M.R. acknowledges useful and interesting discussions with Hanno Gottschalk.

**Funding**

No third-party funding or support has been received for this work.

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
