# A Theoretical Consideration of the One-vs-All Classifier

Let $C(i|x, y)$, $y \in \{1, \ldots, n\}$ be an ensemble of binary classifiers with $C(o|x, y) = 1 - C(i|x, y)$. Furthermore we define the joint distribution

$$p_U(x, y) = p(x|y)p_U(y) \tag{12}$$

w.r.t. a uniform class distribution, i.e., $p_U(y) \equiv \frac{1}{n} \ \forall y \in \mathcal{Y}$. Lastly we define

$$\tilde{C}(i|x, y) = \frac{\frac{1}{n}C(i|x, y)}{\frac{1}{n}C(i|x, y) + \frac{n-1}{n}C(o|x, y)} . \tag{13}$$

Additionally, $\hat{p}(y)$, $\forall y \in \mathcal{Y}$, are the estimated relative class frequencies. Recalling (1), our proposed training objective for the ensemble of binary classifiers is defined by

$$\min_C \frac{1}{|S|} \sum_{(x,y) \in S} \left[ -\log(C(i|x, y)) - \frac{1}{n-1} \sum_{y' \in \mathcal{Y} \setminus \{y\}} \frac{\hat{p}(y)}{\hat{p}(y')} \log(C(o|x, y')) \right] . \tag{1}$$

**Assumption 1.** *We make the following assumptions for our one-vs-all classifier $C$*

1. *We sample $S \sim p(x, y)$ i.i.d. and there is no GAN-generated data involved;*

2. *Let $\mathcal{H} = \{C\}$ the set of all realizable one-vs-all classifiers. We assume, there exists a $C^* \in \mathcal{H}$ such that $\hat{p}(y|x) = p(y|x)$, i.e., $p(y|x)$ is realizable;*

3. *We can compute an empirical risk minimizer, i.e., we can determine a $C_S \in \mathcal{H}$ which minimizes (1) for a given sample $S$.*

Note that the above assumptions are typical assumptions in statistical learning theory [42, 49]. Under these assumptions, the empirical risk minimizer converges to the desired hypothesis $C^*$ for $|S| \to \infty$. Furthermore, (deep) neural networks have an asymptotic (i.e., for increasing network capacity) universal approximation property [54] which makes assumption 2 fairly realistic.

**Lemma 1** (Class posterior). *Under assumption 1 and training $C$ on equation (1) it holds that for $|S| \to \infty$*

$$\hat{p}(y|x) = \frac{\tilde{C}(i|x, y)\hat{p}(y)}{\sum_{y' \in \mathcal{Y}} \tilde{C}(i|x, y')\hat{p}(y')} \longrightarrow p(y|x) . \tag{3}$$

*Proof.* Without loss of generality consider a single one-vs-all classifier $C(i|x, y^*)$ with $y^* \in \mathcal{Y}$ fixed and define $\bar{y^*} := \mathcal{Y} \setminus \{y^*\}$ as the counter-part class of class $y^*$ (class "not $y^*$").
If we now sample $S_{y^*} \sim \tilde{p}_{y^*}(x, y) = p(x|y)\tilde{p}_{y^*}(y)$, with $\tilde{p}_{y^*}(y^*) = \frac{1}{2}$ and $\tilde{p}_{y^*}(y) = \frac{1}{2(n-1)}$, $\forall y \in \mathcal{Y} \setminus \{y^*\}$, we are weighting $y^*$ and $\bar{y^*}$ equally. The loss contribution of $C(i|x, y^*)$ in equation (1) then becomes

$$\frac{1}{|S_{y^*}|} \sum_{(x,y) \in S_{y^*}} -\mathbb{1}_{\{y=y^*\}} \log(C(i|x, y^*)) - \frac{1}{n-1} \mathbb{1}_{\{y \neq y^*\}} \frac{\frac{1}{2}}{\frac{1}{2(n-1)}} \log(C(o|x, y^*)) \tag{14}$$

$$= \frac{1}{|S_{y^*}|} \sum_{(x,y) \in S_{y^*}} -\mathbb{1}_{\{y=y^*\}} \log(C(i|x, y^*)) - \mathbb{1}_{\{y \neq y^*\}} \log(C(o|x, y^*)) , \tag{15}$$

which is the binary cross entropy loss for equal class weights of $y^*$ and $\bar{y^*}$. This shows that our chosen loss function (1) yields a balanced one-vs-all classifier. As the change in sampling from the classes does not affect $p(x|y)$, by assumptions 1–3 we obtain for $|S| \to \infty$ that

$$C(i|x, y) \longrightarrow \frac{p(x|y)}{p(x|y) + p(x|\bar{y})}, \quad \forall y \in \mathcal{Y} . \tag{16}$$

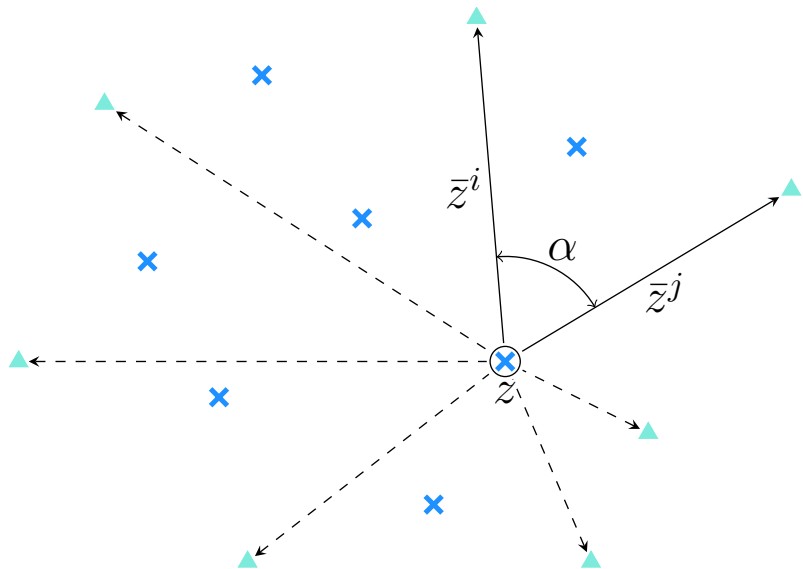

Figure 4: Illustration of the low dimensional regularizer for a single class and example (c.f. equation (10)).

This implies that the re-weighted classifier $\tilde{C}$ for $|S| \to \infty$ fulfills

$$\tilde{C}(i|x,y) = \frac{\frac{1}{n}C(i|x,y)}{\frac{1}{n}C(i|x,y) + \frac{n-1}{n}C(o|x,y)} \longrightarrow \frac{\frac{\frac{1}{n}p(x|y)}{p(x|y)+p(x|\bar{y})}}{\frac{\frac{1}{n}p(x|y)+\frac{n-1}{n}p(x|\bar{y})}{p(x|y)+p(x|\bar{y})}} \tag{17}$$

$$= \frac{\frac{1}{n}p(x|y)}{\frac{1}{n}p(x|y) + \frac{n-1}{n}p(x|\bar{y})} \tag{18}$$

$$= \frac{p(x|y)p_U(y)}{p(x|y)p_U(y) + p(x|\bar{y})p_U(\bar{y})} \tag{19}$$

$$= \frac{p(x|y)p_U(y)}{p_U(x)} \tag{20}$$

$$= p_U(y|x). \tag{21}$$

By the preceding convergence, we obtain

$$\frac{\tilde{C}(i|x,y)\hat{p}(y)}{\sum_{y'}\tilde{C}(i|x,y')\hat{p}(y')} \longrightarrow \frac{p_U(y|x)p(y)}{\sum_{y'}p_U(y'|x)p(y')} \tag{22}$$

$$= \frac{\frac{p_U(x|y)p_U(y)p(y)}{p_U(x)}}{\sum_{y'}\frac{p_U(x|y')p_U(y')p(y')}{p_U(x)}} \tag{23}$$

$$= \frac{p(x|y)p(y)}{\sum_{y'}p(x|y')p(y')} \tag{24}$$

$$= p(y|x), \tag{25}$$

for $|S| \to \infty$, which concludes the proof. $\square$

# B  Low Dimensional Regularizer

In this section we explain how our low dimensional regularizer behaves on a simple example, aiming at providing the reader with a clearer intuition on the role of that regularizer. Recalling (11), the low

dimensional regularizer applied to our GAN training is defined as

$$L_R = \frac{1}{n} \sum_{y \in \mathcal{Y}} \left[ \frac{1}{N_y} \sum_{(x,y) \in S} l_R(A_{\mathrm{enc}}(x,y), y) \right] , \qquad (11)$$

with

$$l_R(z,y) = \frac{2}{N_y^z \cdot (N_y^z - 1)} \cdot \sum_{\substack{\bar{z}^i, \bar{z}^j \in \mathcal{Z}(z,y) \\ i < j}} - \log \left( \arccos \left( \frac{\bar{z}^i * \bar{z}^j}{\|\bar{z}^i\| \cdot \|\bar{z}^j\|} \right) \cdot \frac{1}{\pi} \right) , \qquad (10)$$

where for a given example $(x,y) \in S$, $z = A_{\mathrm{enc}}(x,y)$ is the latent embedding of $x$, $\mathcal{Z}(z,y) = \{\tilde{z} - z | \tilde{z} = G(e,y), e \sim U(0,1)\} = \left\{ \bar{z}^1, \dots, \bar{z}^{N_y^z} \right\}$ are all, by the GAN, generated latent codes but normalized to origin $z$ and $N_y = |\{(x,y')|y' = y\}|$ are the number of examples $x$ with class label $y$.

As the regularizer loss is class conditional let us consider a simple example with a single class. In figure 4 you can see an illustration similar to the left Gaussian in figure 1. In this example the cosine similarity between $\bar{z}^i$ and $\bar{z}^j$ can be computed as

$$\cos(\alpha) = \frac{\bar{z}^i * \bar{z}^j}{\|\bar{z}^i\| \cdot \|\bar{z}^j\|} \in [-1, 1] , \qquad (26)$$

where $\cos(\alpha) = -1$ implies that $\bar{z}^i$ and $\bar{z}^j$ point in opposite directions and $\cos(\alpha) = 1$ implies that they point into the same direction. In order to get a distance measure that is taking values in $[0, 1]$, we compute the angle $\alpha$ in radians and normalize the result by $1/\pi$, i.e.,

$$\frac{\alpha}{\pi} = \arccos \left( \frac{\bar{z}^i * \bar{z}^j}{\|\bar{z}^i\| \cdot \|\bar{z}^j\|} \right) \cdot \frac{1}{\pi} \in [0, 1] . \qquad (27)$$

This is now the same quantity as in equation (10). By maximizing the average angular distance between all unique pairs $\bar{z}^i, \bar{z}^j \in \mathcal{Z}(z,y)$, $i < j$, we are forcing the generator to spread all generated latent codes on the boundary of the distribution represented by the class-specific data example. Transforming these angular distances via a logarithm and averaging over all $(x,y) \in S$, yields our low dimensional regularizer in equation (11).

Figure 5 shows OoC examples generated by GANs who where trained with different weights for the low dimensional regularizer ($\lambda_{\mathrm{reg}}$). Especially in the MNIST 0-4 example one can see that the GAN trained with $\lambda_{\mathrm{reg}} = 32$ produces much more diverse samples compared to no low-dimensional regularization.

## C  Additional Toy Examples

As a more challenging 2D example, we also present a result on the two moons dataset. In figure 6, the results of an experiment with two separable classes is shown. In the top row of the figure, the training data is class-wise separable. In that case, we observe that the decision boundary also belongs to the OoD regime (top left panel), which is true as there is no in-distribution data present. Our model is able to learn this since the classes are shielded tightly enough such that the generated OoC examples are in part also located in the vicinity of the decision boundary. To better visualize the distribution of generated data, we depict estimated densities of the generated OoC data in the top right panel. For the aleatoric uncertainty in the top center panel, we observe that due to numerical issues, aleatoric uncertainty increases further away from the in-distribution data. However this can be accounted for by first considering epistemic uncertainty and then the aleatoric one. By this procedure, most of the examples close to the decision boundary would be correctly classified as OoD which is also correct since there is only a minor amount of aleatoric uncertainty involved in this example due to moderate sample size not being reflected by the data.

In the bottom row example, an experiment analogous to the top row but with a noisier version of the data is presented. The bottom left panel shows that the epistemic uncertainty on the decision boundary between the two classes clearly decreases in comparison to the top left panel. At the same time the bottom center panel shows the gain in aleatoric uncertainty compared to the top center panel. Note that for data points far away from the in-distribution regime, all $C(i|x,y)$ take values close

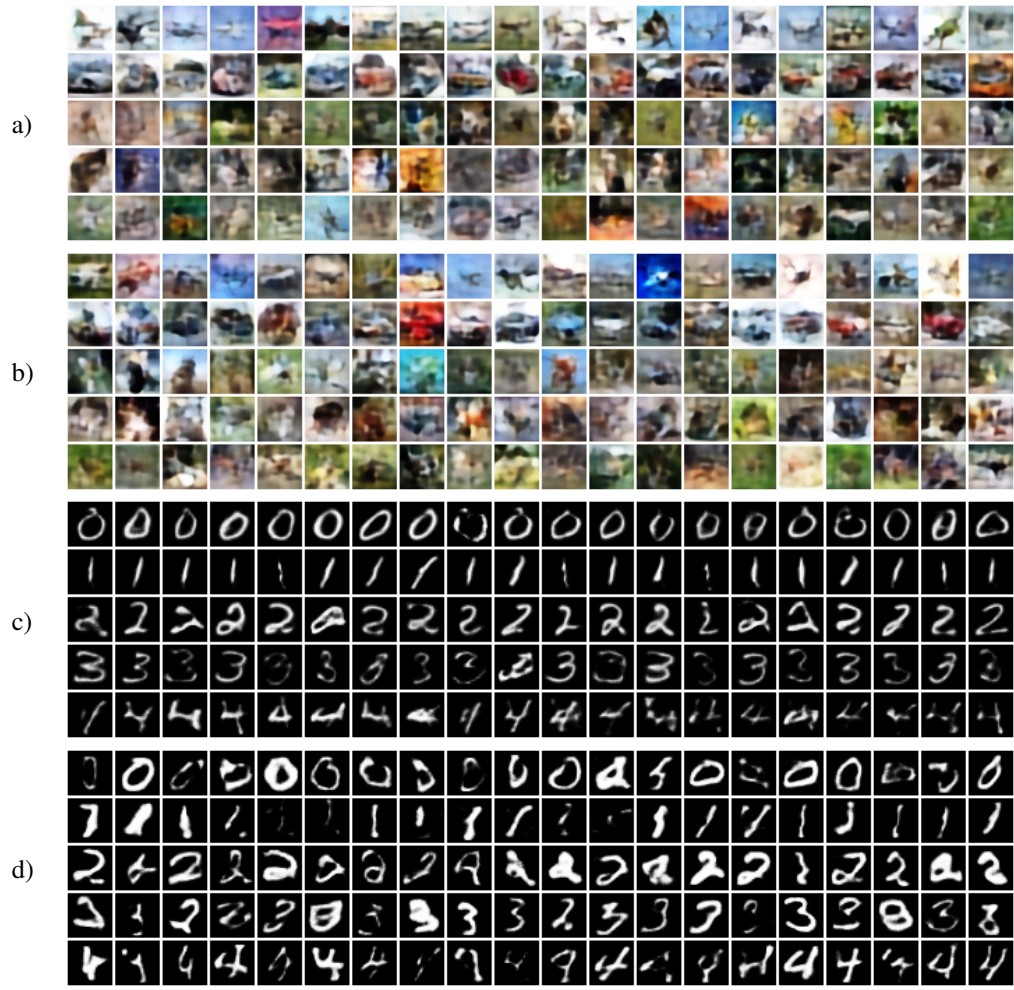

Figure 5: Generated OoC examples by our approach with different weights on the low dimensional regularizer. a) CIFAR10 0-4 samples with $\lambda_{\text{reg}} = 0$; b) CIFAR10 0-4 samples with $\lambda_{\text{reg}} = 1$; c) MNIST 0-4 samples with $\lambda_{\text{reg}} = 0$; d) MNIST 0-4 samples with $\lambda_{\text{reg}} = 32$.

to zero. In practice this requires the inclusion of a small $\varepsilon > 0$ in the denominator to circumvent numerical problems in the logarithmic loss terms. In our experiments this also results in a high aleatoric uncertainty far away from the in-distribution as all estimated probabilities uniformly take the lower bound's value $\varepsilon$. However, a joint consideration of aleatoric and epistemic uncertainty disentangles this, since a high estimated probability of being OoD means that the estimate of aleatoric uncertainty can be neglected. This also becomes evident in the center panels where one can observe high aleatoric uncertainty $H(x)$ outside the in-distribution regime, which can however be masked out by the OoD probability $\tilde{C}(o|x)$.

Figure 7 shows a toy example on a $3 \times 3$ grid of Gaussians. In the top row each Gaussian belongs to its own class, resulting in 9 classes in total, while in the bottom row multiple Gaussians belong to one class, resulting in disconnected class regions. In both cases our method is able to predict a high aleatoric uncertainty between class boundaries and high epistemic uncertainty away from the training data. We can also observe the high aleatoric uncertainty far away from the training data as in figure 6, which can also be disentangled by a joint consideration of aleatoric and epistemic uncertainty.

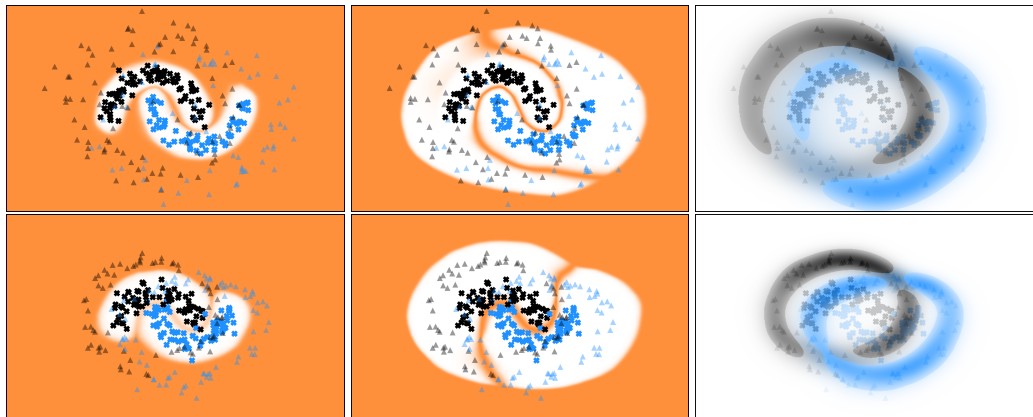

Figure 6: Two toy examples of the two moons dataset with different variance. From left to right: 1. OoD heatmap with orange indicating a high probability of being OoD and white for in-distribution; 2. Aleatoric uncertainty (entropy over Equation (3)) with orange indicating high and white low uncertainty; 3. Gaussian kernel density estimate of the GAN examples. Triangles indicate GAN OoC examples and crosses correspond to the in-distribution data. The data underlying the bottom row has higher variance than the one underlying the top row.

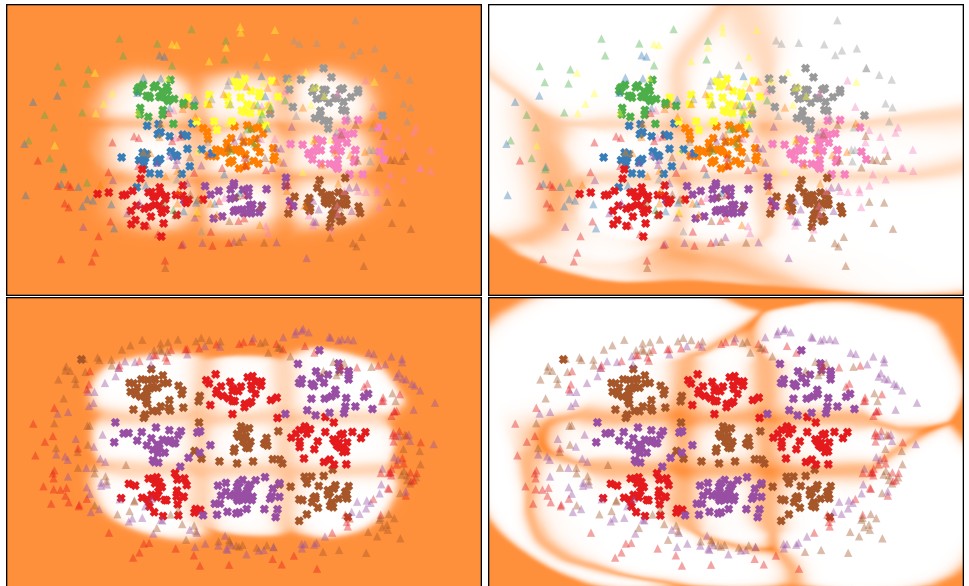

Figure 7: Two toy examples of a 3x3 grid of independent Gaussians. In the top row each Gaussian has its own class assigned, resulting in 9 classes and in the bottom row disconnected Gaussians were assigned to 3 classes in total. From left to right: 1. OoD heatmap with orange indicating a high probability of being OoD and white for in-distribution; 2. Aleatoric uncertainty (entropy over Equation (3)) with orange indicating high and white low uncertainty; Triangles indicate GAN OoC examples and crosses correspond to the in-distribution data.

## D   Hyperparameter Settings for Experiments

For the Bayes-by-Backprop implementation we use *spike-and-slap* priors in combination with diagonal Gaussian posterior distributions as described in [2]. MC-Dropout uses a 50% dropout probability on all weight layers. Both mentioned methods average their predictions over 50 forward passes. The deep-ensembles were built by averaging 5 networks. Implementations of Confident Classifier and GEN use the architectures and hyperparameters recommended by the authors and we followed their reference code where possible. Parameter studies showed that our method is

mostly stable w.r.t. the hyperparameter selection. We used $\lambda_{gp} = 10$ as proposed in [11], $\lambda_{cl} = 2$ for MNIST/Tiny Imagenet and $\lambda_{cl} = 4$ for CIFAR10/100, $\lambda_{\text{real}} = 0.6$, $\lambda_R = 32$ for MNIST/Tiny ImageNet and $\lambda_R = 1$ for CIFAR10/100.

The latent dimension for MNIST was set to 32 while using 128 dimensions for CIFAR10/100 and Tiny ImageNet. We used batch stochastic gradient descent with the ADAM [19] optimizer and a batch size of 256. The learning rate was initialized to $10^{-3}$ for the classification model and $2 \cdot 10^{-4}$ for the GAN while linearly decaying them to $10^{-5}$ over the course of all training iterations. Training on the MNIST dataset required $2\,000$ generator iterations while taking $10\,000$ iterations for the CIFAR10/100 and Tiny ImageNet datasets (one iteration is considered to be one gradient step on the generator). As recommended in [11], we use batch normalization only in the generator, while the critic as well as the classifier do not use any type of layer normalization. We also adopt the alternating training scheme from the just mentioned work. For each generator iteration the critic as well as classifier are performing 5 optimization steps on the same batch. We do apply some mild data augmentation by using random horizontal flipping where appropriate. The test set sizes used for computing the numerical results can be found in table 6. Besides the already mentioned LeNet and ResNet architectures for the classification model, the cAE uses a small convolutional architecture for MNIST and a ResNet architecture for CIFAR10/100 and Tiny ImageNet. The generator and discriminator are both implemented as fully connected networks with $(1024, 512, 256)$ neurons for the generator and $(512, 512, 512)$ neurons for the discriminator. We experimented with different sizes of the generator and observed low to no influence on the final performance of the classifier. All experiments were performed on a Nvidia RTX 3090, Tesla P100 or A100 GPU but models with less VRAM are also sufficient as the cGAN itself is very small. On the RTX 3090 training takes approximately 50 minutes, 3 hours, 3.5 hours and 5.5 hours for MNIST, CIFAR10, CIFAR100 and Tiny ImageNet, respectively.

Table 6: Test set sizes used for computing the numerical results.

| Dataset | Test-Set Size |
|---|---|
| MNIST 0-4 | 5 000 |
| MNIST 5-9 | 5 000 |
| CIFAR10 0-4 | 5 000 |
| CIFAR10 5-9 | 5 000 |
| Tiny ImageNet 0-99 | 5 000 |
| Tiny ImageNet 100-199 | 5 000 |
| EMNIST-Letters | 20 800 |
| Fashion-MNIST | 10 000 |
| SVHN | 26 032 |
| Omniglot | 13 180 |
| LSUN | 10 000 |

# E   Parameter Study

In order to examine the influence of hyperparameter selection onto our framework we conducted an extensive experimental study. We displayed all mentioned evaluation metrics from section 4 while varying $\lambda_{\text{cl}}$, $\lambda_{\text{real}}$, $\lambda_{\text{reg}}$ and the chosen latent dimension for the cAE and cGAN. For $\lambda_{\text{cl}}$ in equation (8), which controls the influence of the classifier predictions onto the generated OoC examples, figure 9 shows clearly that for MNIST $\lambda_{\text{cl}} = 2$ and for CIFAR10 $\lambda_{\text{cl}} = 4$ are locally optimal values w.r.t. maximum performance. While larger $\lambda_{\text{cl}}$ tend to increase the in-distribution accuracy slightly it greatly decreases all other evaluation metrics. A very interesting observation in figure 10 about the influence of $\lambda_{\text{real}}$ from equation (9) is that both extremes ($\lambda_{\text{real}} \in \{0, 1\}$) are greatly decreasing the results. This shows clearly that the generated OoC examples have a positive effect on the OoD detection performance and in-distribution separability. In terms of the dimensionality of the latent space, figure 11 shows that 32 and 128 dimensions are the optimal values for MNIST and CIFAR10, respectively. This is coherent with the visual quality of the examples decoded by the cAE, which does not improve much with higher dimensions. Analysing the influence of $\lambda_{\text{reg}}$ in figure 12 one can observe a positive effect on the results on the MNIST dataset by increasing the value of the parameter up to $\lambda_{\text{reg}} = 32$ where we reach a local maximum. It is also very apparent, that for $\lambda_{\text{reg}} = 0$ the results are comparatively bad, emphasizing the role of the low dimensional regularizer in our model. For CIFAR10 the effect is not as clear as for the MNIST dataset, but figure 12 also shows a locally

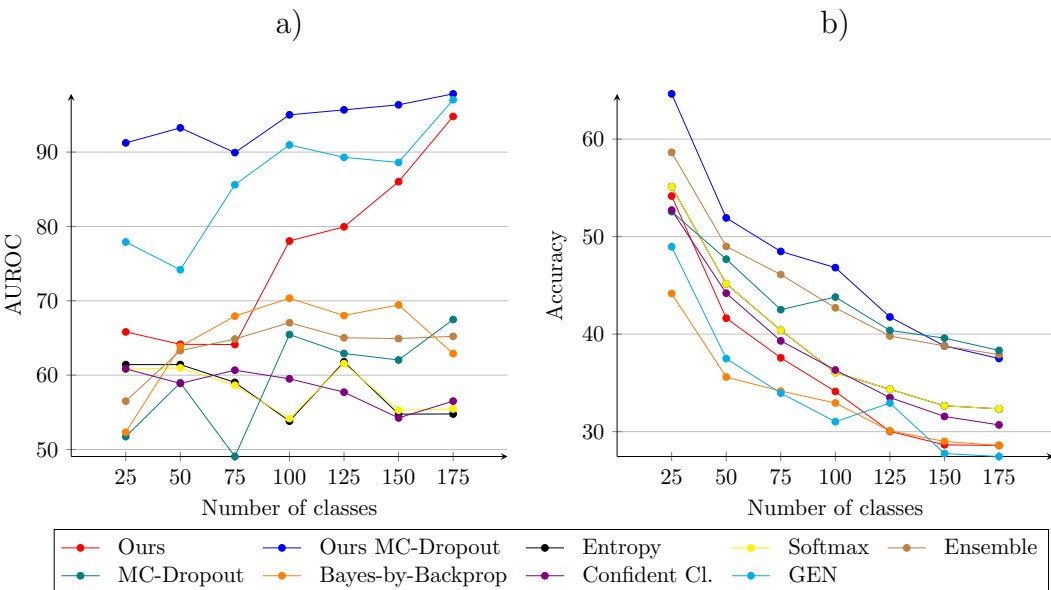

Figure 8: Influence of the number of classes on the OoD detection and accuracy based on the Tiny ImageNet dataset. a) Influence on the AUROC; b) Influence on the accuracy. The number of classes corresponds to the Tiny ImageNet dataset and the rest of the classes are then considered as OoD. Additonally the SVHN, Fashion-MNIST and MNIST datasets were used as OoD. Hyperparameters and models are the same as for the results in tables 5 and 11 and kept fixed for different numbers of classes.

optimal setting of $\lambda_{\text{reg}} \in [1, 2]$. We believe that the higher latent dimension required for the CIFAR10 dataset and thus the curse of dimensionality is the main factor behind this finding. Figure 5 shows qualitatively the influence of $\lambda_{\text{reg}}$ on the generated OoC examples. We can observe that for a training with a higher $\lambda_{\text{reg}}$ we obtain much more diverse examples compared to when using no regularizer. This is especially apparent in the MNIST setting.

Figure 8 shows the OoD detection performance w.r.t. AUROC and the accuracy on the in-distribution set while increasing the number of classes present during training. The hyperparameters as well as all model architectures were kept fixed for this experiment. We used the Tiny ImageNet dataset with varying class-wise subsets as in-distribution data and the remaining Tiny ImageNet classes, SVHN, Fashion-MNIST and MNIST datasets as OoD examples. All methods were trained on our training subset while the figure displays the results on the respective test sets. For GEN we faced numerical problems with exploding evidence values. Due to this, we chose different hyperparameter settings for different numbers of classes, which also results in larger standard deviation values for GEN in tables 5 and 11. The study shows that increasing the number of classes also increases the OoD detection performance of our approach. This is also true for GEN, which is why we argue that the gain in performance can in part be attributed to the use of an autoencoder. As the number of classes increases, so does the number of examples in the training dataset, improving the ability of the autoencoder to produce diverse embeddings. When observing the accuracy on the in-distribution data the most apparent observation is that for all methods the accuracy decreases for a higher number of classes. This is not surprising as we kept the classification model fixed. In general we observe that our approach with MC-dropout has the overall highest performance, being equal to standard MC-Dropout and Ensembles from $150$ classes and above. Our approach without dropout achieves mid-tier results while still being superior to GEN.

## F  Joint Detection of OoD and FP in the Wild

In this section, we perform CIFAR10-based OoD and FP detection in the wild, i.e., we perform both tasks jointly while presenting in-distribution and OoD data in the same mix as in table 3 to the

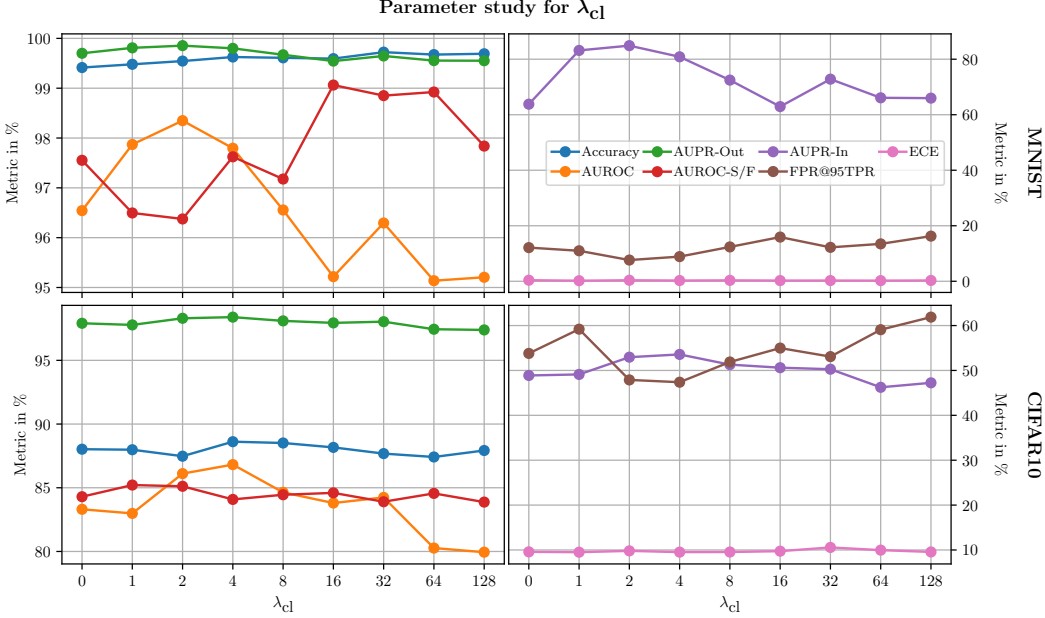

Figure 9: Parameter study over $\lambda_{cl}$ in equation (8). For MNIST hyperparameters were fixed at $\lambda_{reg} = 14$, $\lambda_{real} = 0.5$, latent dimension = 16 and for CIFAR10 at $\lambda_{reg} = 0$, $\lambda_{real} = 0.6$, latent dimension = 128. All seeds were also the same for all experiments. All metrics were computed on the validation sets of MNIST 0-4 / CIFAR10 0-4 as in-distribution datasets and the entirety of all assigned OoD datasets as defined in section 4.

Table 7: Results obtained from aggregating predicted uncertainty estimates in a gradient boosting model which was then trained on a validation set. In distribution dataset is CIFAR10 (0-4) and OoD datasets are CIFAR10 (5-9), LSUN, SVHN, Fashion-MNIST, MNIST.

| Method | Uncertainty Scores and Predicted Probabilities | | | Uncertainty Scores Only | | |
| --- | --- | --- | --- | --- | --- | --- |
| | Accuracy TP | Accuracy FP | Accuracy OoD | Accuracy TP | Accuracy FP | Accuracy OoD |
| Ours | 78.32 (1.20) | 37.33 (3.05) | 75.47 (0.92) | 76.96 (1.14) | 40.88 (2.65) | 66.07 (0.99) |
| Ours with MC-Dropout | **79.59 (0.61)** | 47.58 (2.68) | **82.50 (0.55)** | **77.16 (1.19)** | 41.86 (2.92) | **70.37 (0.69)** |
| One-vs-All Baseline | 68.54 (1.65) | 46.54 (1.92) | 73.69 (1.81) | 64.78 (2.25) | 33.75 (6.83) | 56.02 (2.61) |
| Max. Softmax [15] | 66.04 (0.33) | 49.51 (1.69) | 71.55 (1.48) | 64.98 (0.58) | 33.58 (4.01) | 48.77 (4.34) |
| Entropy | 66.98 (0.56) | 48.47 (1.74) | 72.36 (1.33) | 66.13 (1.07) | 29.90 (5.67) | 51.42 (4.62) |
| Bayes-by-Backprop [2] | 67.98 (0.88) | 44.65 (2.05) | 73.34 (0.99) | 64.81 (0.93) | 33.62 (4.31) | 52.76 (4.31) |
| MC-Dropout [7] | 72.20 (0.41) | **52.53 (1.29)** | 77.38 (0.49) | 66.97 (1.88) | 44.31 (2.14) | 59.91 (2.10) |
| Deep-Ensembles [24] | 72.10 (0.33) | 47.76 (0.90) | 75.30 (0.41) | 67.76 (0.56) | 36.29 (1.78) | 53.56 (1.01) |
| Confident Classifier [27] | 68.15 (0.77) | 50.64 (1.45) | 72.23 (0.71) | 64.69 (1.47) | 39.84 (3.85) | 45.62 (3.47) |
| GEN [40] | 69.86 (2.11) | 49.13 (2.43) | 76.04 (1.17) | 70.47 (0.94) | **64.88 (1.52)** | 54.31 (2.52) |
| Entropy Oracle | 76.89 (0.32) | 54.33 (2.05) | 86.59 (0.37) | 71.36 (1.59) | 61.40 (2.77) | 82.35 (0.62) |
| One-vs-All Oracle | 75.35 (0.90) | 58.20 (2.03) | 80.00 (2.43) | 73.60 (1.74) | 60.49 (0.84) | 74.78 (2.85) |

classifier. To this end, we apply gradient boosting to the uncertainty scores provided by the respective methods to predict TP, FP and OoD. In more detail, we use the following uncertainty scores:

- Ours: OoD uncertainty $C(o|x)$ and entropy $H(x)$ of the estimated class probabilites
- Ours with MC dropout: $C(o|x)$, $H(x)$ and the standard deviations of $\hat{p}(y|x)$ summed over all $y = 1, \ldots, n$ under MC dropout for 50 forward passes.
- One-vs-All Baseline: Same as "Ours".
- Max softmax: Maximum softmax probability.
- Entropy: Entropy over estimated class probabilities.
- Bayes-by-Backprop: For 50 samples from the posterior we compute $a = \frac{1}{K} \sum_1^K H(\hat{p}(y|x))$ (aleatoric uncertainty) and $b = H(\frac{1}{K} \sum_1^K \hat{p}(y|x)) - a$ (epistemic uncertainty) as in [18].
- MC-Dropout: Entropy of estimated class probabilities averaged over 50 forward passes, standard deviation of the class probabilities summed over all $y = 1, \ldots, n$.

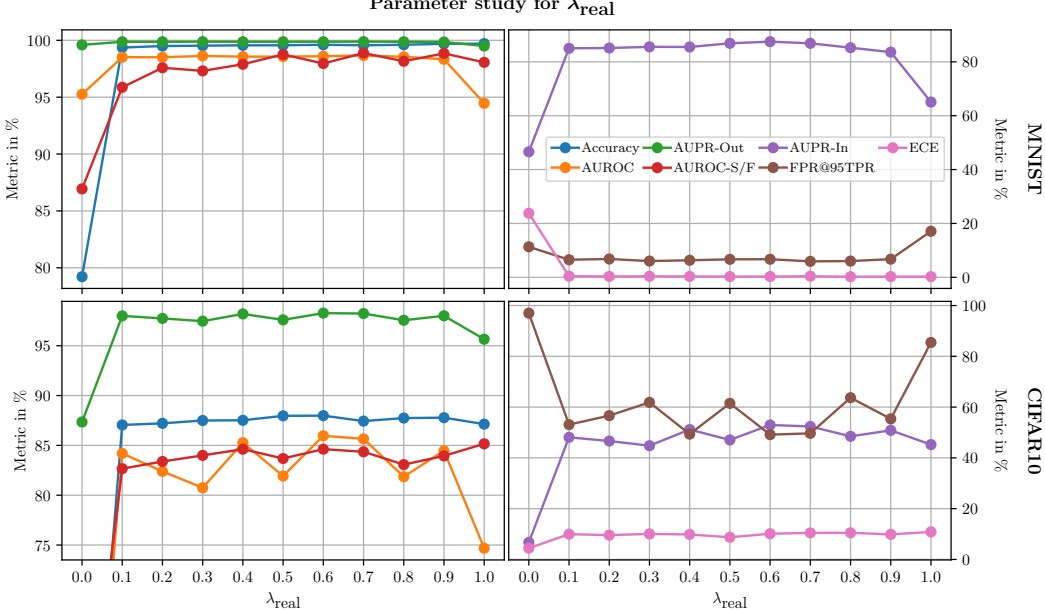

Figure 10: Parameter study over $\lambda_{\text{real}}$ in equation (9). For MNIST, hyperparameters were fixed at $\lambda_{\text{reg}} = 32$, $\lambda_{\text{cl}} = 1$, latent dimension $= 16$ and for CIFAR10 at $\lambda_{\text{reg}} = 0$, $\lambda_{\text{cl}} = 2$, latent dimension $= 128$. All seeds were also the same for all experiments. All metrics were computed on the validation sets of MNIST 0-4 / CIFAR10 0-4 as in-distribution datasets and the entirety of all assigned OoD datasets as defined in section 4.

- Deep-Ensembles: Entropy of estimated class probabilites averaged over 5 ensemble members, standard deviation of the class probabilities summed over all $y = 1, \ldots, n$.
- Confident Classifier: Entropy over estimated class probabilities.
- GEN: Entropy over estimated class probabilities resulting of the estimated evidence of the Dirichlet distribution.

The corresponding results are given in table 7 in terms of class-wise accuracy. While the right-hand half of table 7 presents results for gradient boosting applied to the uncertainty scores of each method, aiming to predict TP, FP and OoD, the left half of the table shows analogous results while additionally using the estimated class probabilities $\hat{p}(y|x)$ as inputs for gradient boosting. We do so for the sake of accounting for other possible transformations of $\hat{p}(y|x)$ that are not explicitly constructed. The main observations are that our method outperforms the other GAN-based methods and that our method including dropout achieves the overall best performance. It can be observed that the Entropy Oracle performs very strong while using only a single uncertainty score. At a second glance, this is not surprising since a FP mostly involves the confusion of two classes while training the DNN to output maximal entropy on OoD examples is likely to result in the confusion of up to five classes, therefore yielding different entropy levels. Also in the left part as well as the right part of the table, our method including dropout is fairly close the best oracle, which is the entropy oracle. Apart from the oracle, in both studies including and excluding the estimated class probabilities $\hat{p}(y|x)$, our method including MC-dropout outperforms all other methods. However, reviewing the result in an absolute sense, there still remains plenty of room for improvement.

# G   Detailed Results on Individual OoD Datasets

In section 4 we have presented results on the in-distribution datasets MNIST 0-4, CIFAR10 0-4 and CIFAR100 0-49 versus the entirety of all respective OoD datasets. To give more detailed insights, this appendix section contains comparisons of the respective in-distribution dataset and each of the single corresponding OoD datasets.

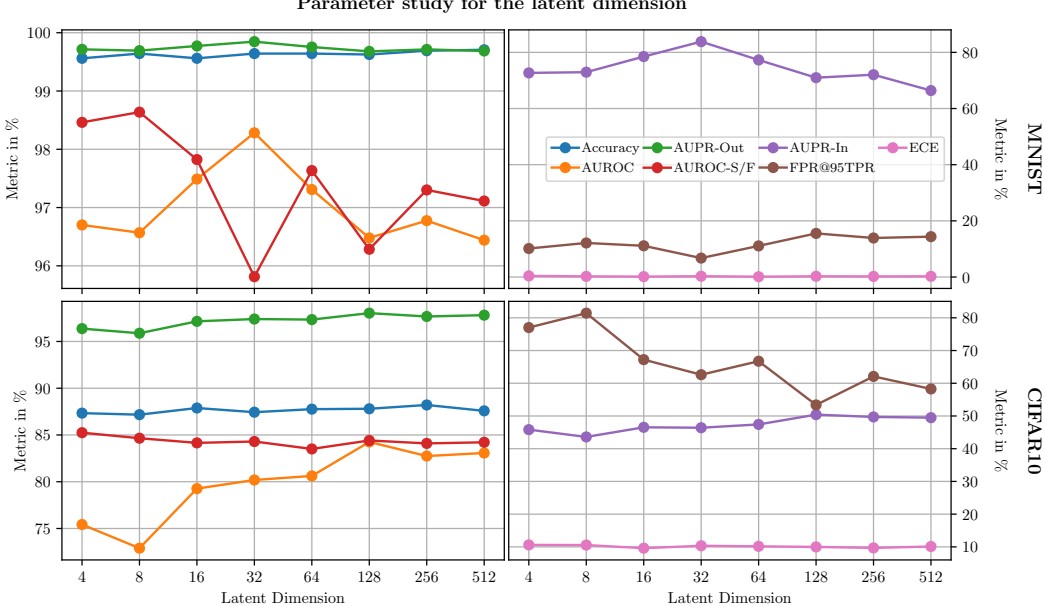

Figure 11: Parameter study over latent dimensions of $z$ in equation (8). For MNIST, hyperparameters were fixed at $\lambda_{\text{reg}} = 32$, $\lambda_{\text{cl}} = 1$, $\lambda_{\text{real}} = 0.5$ and for CIFAR10 at $\lambda_{\text{reg}} = 0$, $\lambda_{\text{cl}} = 4$, $\lambda_{\text{real}} = 0.6$. All seeds were also the same for all experiments. All metrics were computed on the validation sets of MNIST 0-4 / CIFAR10 0-4 as in-distribution datasets and the entirety of all assigned OoD datasets as defined in section 4.

The comparison by dataset for the MNIST 0-4 as in-distribution set in the OoD-dataset-wise break-down given in table 8 shows that we achieve mid-tier results compared to the other methods, except for when MNIST 5-9 is considered as the OoD dataset. In that challenging case, we achieve stronger results compared to the other methods. Examining the results of our OoD experiments with CIFAR10 0-4 being the in-distribution dataset in table 9, it becomes apparent that we consistently outperform all other methods on each single OoD dataset. The performance gains are particularly pronounced when using the very similar datasets CIFAR10 5-9 and LSUN as OoD datasets. For CIFAR100 0-49 we have a similar situation as for MNIST, where our approach performs particularly well on the difficult task of CIFAR100 0-49 vs. CIFAR100 50-99 while achieving mid-tier results on the other tasks. This consistency supports the finding that our method – especially in the case of OoD datasets very similar to the in-distribution dataset – shows the most improvement compared to other methods. This finding might be to some extent attributable to our tight class shielding.

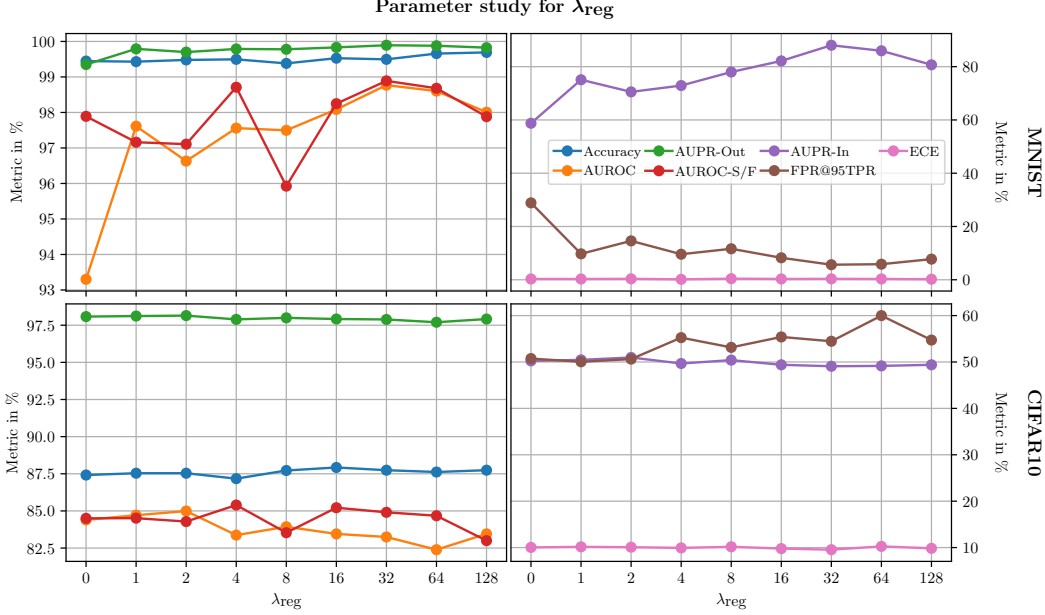

Figure 12: Parameter study over $\lambda_{\mathrm{reg}}$ in equation (8). For MNIST, hyperparameters were fixed at $\lambda_{\mathrm{cl}} = 1$, $\lambda_{\mathrm{real}} = 0.5$, latent dimension $= 16$ and for CIFAR10 at $\lambda_{\mathrm{cl}} = 4$, $\lambda_{\mathrm{real}} = 0.6$, latent dimension $= 128$. All seeds were also the same for all experiments. All metrics were computed on the validation sets of MNIST 0-4 / CIFAR10 0-4 as in-distribution datasets and the entirety of all assigned OoD datasets as defined in section 4.

Table 8: An OoD-dataset-wise breakdown of the results given in table 2.

| Method | AUROC ↑ | AUPR-In ↑ | AUPR-Out ↑ | FPR@ 95% TPR ↓ | AUROC ↑ | AUPR-In ↑ | AUPR-Out ↑ | FPR@ 95% TPR ↓ |
|---|---|---|---|---|---|---|---|---|
| | MNIST 0-4 vs. MNIST 5-9 | | | | MNIST 0-4 vs. EMNIST-Letters | | | |
| Ours | 92.34 (2.31) | 93.17 (2.78) | 89.96 (2.23) | 38.68 (7.09) | 96.44 (0.76) | 86.32 (2.69) | 99.11 (0.20) | 13.80 (3.06) |
| Ours + Dropout | **95.50 (0.53)** | **96.22 (0.59)** | 93.66 (0.83) | 24.46 (2.20) | 96.71 (0.55) | 86.99 (2.49) | 99.18 (0.14) | 12.49 (1.63) |
| One-vs-All Baseline | 93.65 (0.97) | 92.32 (1.36) | 94.36 (0.84) | 20.06 (2.98) | 91.32 (0.31) | 70.45 (1.43) | 97.69 (0.10) | 29.30 (1.12) |
| Max. Softmax | 92.77 (0.55) | 91.46 (1.05) | 93.58 (0.26) | 22.17 (1.04) | 91.63 (0.28) | 73.52 (1.02) | 97.72 (0.07) | 29.31 (0.88) |
| Entropy | 92.80 (0.54) | 91.20 (1.27) | 93.63 (0.24) | 22.11 (1.04) | 91.68 (0.27) | 73.51 (1.14) | 97.73 (0.07) | 29.23 (0.91) |
| Bayes-by-Backprop | 93.73 (0.99) | 92.74 (1.58) | 93.83 (0.69) | 22.17 (1.40) | 90.59 (0.80) | 71.60 (2.22) | 97.34 (0.23) | 34.28 (1.54) |
| MC-Dropout | 94.32 (1.10) | 93.05 (1.79) | **95.19 (0.70)** | 17.27 (1.60) | 92.80 (0.37) | 76.85 (1.53) | 98.09 (0.09) | 26.72 (1.00) |
| Deep-Ensembles | 94.20 (0.24) | 92.82 (0.18) | 95.00 (0.28) | **16.89 (1.09)** | 93.08 (0.10) | 77.32 (0.73) | 98.16 (0.02) | 24.65 (0.37) |
| Confident Classifier | 95.33 (0.74) | 95.43 (1.00) | 94.95 (0.64) | 19.74 (1.46) | 94.60 (0.41) | 81.71 (1.56) | 98.53 (0.11) | 22.28 (1.87) |
| GEN | 88.91 (2.64) | 86.09 (4.55) | 88.15 (2.20) | 40.49 (5.53) | **99.27 (0.30)** | **97.05 (1.22)** | **99.82 (0.07)** | **3.61 (1.48)** |
| Entropy Oracle | 99.76 (0.04) | 99.77 (0.03) | 99.74 (0.04) | 0.93 (0.12) | 99.84 (0.04) | 99.43 (0.12) | 99.96 (0.01) | 0.73 (0.21) |
| One-vs-All Oracle | 99.65 (0.04) | 99.66 (0.03) | 99.65 (0.04) | 1.21 (0.13) | 99.87 (0.01) | 99.49 (0.05) | 99.97 (0.00) | 0.59 (0.10) |
| | MNIST 0-4 vs. Omniglot | | | | MNIST 0-4 vs. Fashion-MNIST | | | |
| Ours | 96.70 (0.52) | 94.25 (1.07) | 98.32 (0.26) | 18.00 (3.26) | 99.36 (0.19) | 99.04 (0.32) | 99.62 (0.10) | 2.19 (1.36) |
| Ours + Dropout | 98.45 (0.24) | 96.94 (0.37) | 99.25 (0.14) | 5.95 (1.08) | 99.65 (0.24) | 99.52 (0.35) | 99.78 (0.17) | 0.68 (0.98) |
| One-vs-All Baseline | 98.75 (0.11) | 97.60 (0.22) | 99.41 (0.05) | 4.75 (0.74) | 99.39 (0.12) | 99.15 (0.16) | 99.59 (0.10) | 1.60 (0.53) |
| Max. Softmax | 98.54 (0.06) | 97.10 (0.24) | 99.31 (0.02) | 5.52 (0.50) | 99.29 (0.23) | 99.03 (0.36) | 99.53 (0.15) | 1.79 (1.34) |
| Entropy | 98.59 (0.06) | 97.15 (0.24) | 99.35 (0.01) | 5.36 (0.50) | 99.35 (0.22) | 99.06 (0.36) | 99.59 (0.15) | 1.74 (1.31) |
| Bayes-by-Backprop | 96.82 (0.16) | 95.22 (0.30) | 97.46 (0.23) | 14.86 (0.56) | 97.93 (0.18) | 97.64 (0.24) | 98.26 (0.20) | 8.02 (1.30) |
| MC-Dropout | **98.96 (0.09)** | 97.56 (0.27) | **99.58 (0.03)** | 3.85 (0.43) | 99.69 (0.05) | 99.52 (0.08) | 99.83 (0.03) | 0.75 (0.21) |
| Deep-Ensembles | 98.93 (0.04) | **97.77 (0.07)** | 99.52 (0.02) | **3.76 (0.20)** | 99.59 (0.09) | 99.38 (0.15) | 99.76 (0.05) | 1.01 (0.63) |
| Confident Classifier | 98.35 (0.29) | 96.22 (0.83) | 99.29 (0.11) | 6.78 (1.06) | **99.95 (0.01)** | **99.91 (0.03)** | **99.97 (0.01)** | **0.06 (0.02)** |
| GEN | 91.03 (3.53) | 78.34 (9.87) | 94.92 (1.67) | 36.96 (9.71) | 99.92 (0.02) | 99.84 (0.04) | 99.96 (0.01) | 0.23 (0.11) |
| Entropy Oracle | 99.68 (0.09) | 99.25 (0.21) | 99.87 (0.04) | 1.26 (0.35) | 100.00 (0.00) | 100.00 (0.00) | 100.00 (0.00) | 0.00 (0.00) |
| One-vs-All Oracle | 99.68 (0.05) | 99.18 (0.13) | 99.88 (0.02) | 1.22 (0.17) | 100.00 (0.00) | 100.00 (0.00) | 100.00 (0.00) | 0.00 (0.00) |
| | MNIST 0-4 vs. SVHN | | | | MNIST 0-4 vs. CIFAR10 | | | |
| Ours | 99.83 (0.09) | 99.50 (0.21) | 99.96 (0.02) | 0.17 (0.06) | 99.84 (0.08) | 99.76 (0.05) | 99.91 (0.05) | 0.21 (0.12) |
| Ours + Dropout | 99.80 (0.22) | 99.47 (0.53) | 99.94 (0.07) | 0.32 (0.40) | 99.84 (0.19) | 99.78 (0.26) | 99.90 (0.12) | 0.28 (0.47) |
| One-vs-All Baseline | 99.76 (0.07) | 99.30 (0.17) | 99.93 (0.04) | 0.36 (0.12) | 99.55 (0.16) | 99.39 (0.21) | 99.69 (0.14) | 0.75 (0.45) |
| Max. Softmax | 99.68 (0.12) | 99.14 (0.24) | 99.92 (0.04) | 0.39 (0.12) | 99.54 (0.13) | 99.39 (0.17) | 99.69 (0.10) | 0.59 (0.30) |
| Entropy | 99.75 (0.11) | 99.22 (0.23) | 99.94 (0.03) | 0.37 (0.11) | 99.61 (0.13) | 99.45 (0.17) | 99.76 (0.09) | 0.56 (0.29) |
| Bayes-by-Backprop | 97.20 (0.32) | 95.75 (0.34) | 98.77 (0.24) | 11.46 (3.55) | 97.52 (0.40) | 97.48 (0.35) | 97.11 (0.73) | 7.44 (2.13) |
| MC-Dropout | 99.94 (0.01) | 99.75 (0.05) | 99.99 (0.00) | 0.15 (0.03) | 99.92 (0.03) | 99.87 (0.05) | 99.96 (0.02) | 0.09 (0.05) |
| Deep-Ensembles | 99.89 (0.01) | 99.55 (0.05) | 99.98 (0.00) | 0.25 (0.06) | 99.82 (0.05) | 99.73 (0.07) | 99.90 (0.03) | 0.20 (0.09) |
| Confident Classifier | **100.00 (0.00)** | **100.00 (0.00)** | **100.00 (0.00)** | **0.00 (0.00)** | **100.00 (0.00)** | **100.00 (0.00)** | **100.00 (0.00)** | **0.00 (0.00)** |
| GEN | 100.00 (0.00) | 100.00 (0.00) | 100.00 (0.00) | 0.00 (0.00) | 100.00 (0.00) | 100.00 (0.01) | 100.00 (0.00) | 0.01 (0.01) |
| Entropy Oracle | 100.00 (0.00) | 100.00 (0.00) | 100.00 (0.00) | 0.00 (0.00) | 100.00 (0.00) | 100.00 (0.00) | 100.00 (0.00) | 0.00 (0.00) |
| One-vs-All Oracle | 100.00 (0.00) | 100.00 (0.00) | 100.00 (0.00) | 0.00 (0.00) | 100.00 (0.00) | 100.00 (0.00) | 100.00 (0.00) | 0.00 (0.00) |

Table 9: An OoD-dataset-wise breakdown of the results given in table 3.

| Method | AUROC ↑ | AUPR-In ↑ | AUPR-Out ↑ | FPR@95% TPR ↓ | AUROC ↑ | AUPR-In ↑ | AUPR-Out ↑ | FPR@95% TPR ↓ |
|---|---|---|---|---|---|---|---|---|
| | CIFAR10 0-4 vs. CIFAR10 5-9 | | | | CIFAR10 0-4 vs. LSUN | | | |
| Ours | 65.94 (0.29) | **72.26 (0.30)** | 64.54 (0.44) | 86.94 (1.12) | 76.45 (0.52) | 69.35 (0.56) | 84.35 (0.42) | 80.50 (1.31) |
| Ours + Dropout | **71.31 (0.49)** | 71.98 (0.35) | **68.45 (0.59)** | 84.07 (0.76) | **82.52 (0.72)** | **76.03 (0.80)** | **88.01 (0.53)** | **72.93 (1.14)** |
| One-vs-All Baseline | 65.31 (0.67) | 66.28 (0.76) | 62.99 (0.44) | 88.15 (0.52) | 74.94 (0.79) | 66.33 (1.06) | 82.99 (0.61) | 82.50 (1.03) |
| Max. Softmax | 64.45 (0.55) | 65.94 (0.80) | 60.77 (0.55) | 90.73 (0.33) | 72.84 (0.59) | 63.69 (0.93) | 80.70 (0.45) | 87.11 (0.56) |
| Entropy | 64.64 (0.53) | 65.82 (0.69) | 61.36 (0.50) | 89.50 (0.69) | 73.33 (0.51) | 63.95 (0.90) | 81.62 (0.39) | 83.58 (0.56) |
| Bayes-by-Backprop | 66.78 (0.34) | 67.16 (1.07) | 63.22 (0.50) | 88.50 (0.62) | 75.31 (0.65) | 66.79 (1.13) | 82.75 (0.68) | 82.64 (1.01) |
| MC-Dropout | 63.52 (0.33) | 64.11 (0.38) | 60.82 (0.27) | 90.01 (0.39) | 77.04 (0.22) | 70.27 (0.41) | 83.75 (0.12) | 81.53 (0.59) |
| Deep-Ensembles | 66.85 (0.38) | 67.64 (0.57) | 64.07 (0.31) | 87.59 (0.42) | 78.03 (0.24) | 69.56 (0.47) | 85.49 (0.20) | 77.07 (0.66) |
| Confident Classifier | 65.58 (0.13) | 66.94 (0.18) | 62.60 (0.29) | 88.46 (0.64) | 75.25 (0.33) | 67.26 (0.41) | 82.97 (0.36) | 81.66 (0.99) |
| GEN | 65.56 (0.50) | 66.67 (0.51) | 61.90 (0.67) | 89.09 (1.17) | 75.82 (1.22) | 67.74 (1.65) | 82.66 (0.75) | 83.19 (1.91) |
| Entropy Oracle | 73.21 (0.52) | 75.00 (0.41) | 70.70 (0.73) | 81.41 (0.88) | 98.27 (0.09) | 96.81 (0.14) | 99.11 (0.06) | 7.91 (0.45) |
| One-vs-All Oracle | 69.45 (0.73) | 69.58 (0.66) | 67.38 (0.80) | 84.92 (0.92) | 95.92 (0.38) | 92.38 (0.65) | 97.93 (0.22) | 19.36 (2.03) |
| | CIFAR10 0-4 vs. SVHN | | | | CIFAR10 0-4 vs. Fashion-MNIST | | | |
| Ours | 98.50 (0.26) | 92.05 (1.07) | 99.72 (0.05) | 5.99 (1.27) | 78.78 (0.88) | 72.41 (0.85) | 85.12 (0.92) | 81.09 (2.84) |
| Ours + Dropout | **98.93 (0.12)** | **94.40 (0.62)** | **99.80 (0.02)** | 5.09 (0.44) | **84.75 (1.32)** | **80.48 (1.45)** | **88.20 (1.38)** | **76.11 (5.07)** |
| One-vs-All Baseline | 70.07 (4.23) | 45.76 (4.99) | 89.65 (1.92) | 91.83 (3.14) | 73.94 (1.31) | 68.28 (1.45) | 80.23 (1.31) | 89.25 (1.33) |
| Max. Softmax | 70.96 (1.87) | 44.63 (2.22) | 90.72 (0.66) | 88.76 (1.38) | 73.90 (1.18) | 67.14 (1.35) | 80.38 (0.97) | 88.48 (0.82) |
| Entropy | 71.23 (1.94) | 44.81 (2.17) | 90.87 (0.72) | 87.04 (2.22) | 73.93 (1.26) | 67.18 (1.38) | 80.30 (1.11) | 88.47 (1.71) |
| Bayes-by-Backprop | 76.21 (0.58) | 50.10 (1.76) | 92.92 (0.24) | 80.85 (1.42) | 74.51 (1.34) | 68.52 (1.75) | 80.67 (1.13) | 88.11 (1.87) |
| MC-Dropout | 76.73 (2.77) | 58.97 (4.10) | 92.23 (0.86) | 84.98 (1.87) | 81.85 (0.75) | 77.62 (0.79) | 85.83 (0.62) | 80.75 (1.85) |
| Deep-Ensembles | 72.02 (1.11) | 45.95 (2.24) | 90.96 (0.30) | 88.13 (0.61) | 72.82 (1.20) | 66.38 (2.02) | 79.25 (0.63) | 89.97 (0.84) |
| Confident Classifier | 73.60 (0.61) | 48.68 (1.08) | 91.71 (0.23) | 85.45 (1.08) | 74.47 (0.52) | 68.40 (0.77) | 80.85 (0.35) | 87.47 (0.60) |
| GEN | 98.43 (0.42) | 91.36 (1.82) | 99.71 (0.08) | 5.62 (1.68) | 72.44 (3.94) | 66.54 (4.86) | 79.12 (2.47) | 89.38 (2.57) |
| Entropy Oracle | 96.85 (0.62) | 88.28 (1.42) | 99.29 (0.21) | 14.26 (2.88) | 97.89 (0.16) | 96.65 (0.28) | 98.71 (0.10) | 9.50 (0.61) |
| One-vs-All Oracle | 89.47 (1.47) | 70.44 (2.54) | 97.36 (0.51) | 48.76 (7.42) | 96.63 (0.40) | 94.34 (0.71) | 98.06 (0.22) | 17.37 (2.48) |
| | CIFAR10 0-4 vs. MNIST | | | | | | | |
| Ours | 83.24 (3.79) | 75.49 (4.07) | 90.43 (2.76) | 58.75 (13.88) | | | | |
| Ours + Dropout | 86.64 (1.18) | **82.03 (1.28)** | 91.49 (1.02) | 61.32 (5.75) | | | | |
| One-vs-All Baseline | 78.65 (0.75) | 75.69 (0.57) | 83.12 (1.09) | 86.70 (2.27) | | | | |
| Max. Softmax | 78.89 (1.79) | 74.29 (2.40) | 84.41 (1.37) | 83.07 (2.10) | | | | |
| Entropy | 79.59 (1.81) | 74.66 (2.38) | 85.33 (1.55) | 77.57 (3.12) | | | | |
| Bayes-by-Backprop | 71.46 (4.50) | 62.19 (6.23) | 79.41 (3.14) | 87.01 (3.35) | | | | |
| MC-Dropout | 82.98 (1.21) | 79.05 (1.68) | 87.44 (0.83) | 73.99 (1.97) | | | | |
| Deep-Ensembles | 81.33 (1.54) | 76.97 (1.67) | 85.98 (1.26) | 78.85 (3.47) | | | | |
| Confident Classifier | 73.41 (2.43) | 64.90 (4.16) | 81.47 (1.62) | 83.21 (2.58) | | | | |
| GEN | **87.63 (3.97)** | 80.69 (5.53) | **93.31 (2.29)** | **45.16 (11.84)** | | | | |
| Entropy Oracle | 97.59 (0.26) | 97.10 (0.30) | 97.82 (0.30) | 10.16 (2.03) | | | | |
| One-vs-All Oracle | 97.56 (0.20) | 96.52 (0.40) | 98.36 (0.07) | 13.25 (1.95) | | | | |

Table 10: An OoD-dataset-wise breakdown of the results given in table 4.

| Method | AUROC ↑ | AUPR-In ↑ | AUPR-Out ↑ | FPR@95% TPR ↓ | AUROC ↑ | AUPR-In ↑ | AUPR-Out ↑ | FPR@95% TPR ↓ |
|---|---|---|---|---|---|---|---|---|
| | CIFAR100 0-49 vs. CIFAR100 50-99 | | | | CIFAR100 0-49 vs. LSUN | | | |
| Ours | 64.52 (0.17) | 65.00 (0.35) | 61.84 (0.21) | 89.57 (0.47) | 65.30 (0.56) | 52.94 (0.30) | 75.99 (0.42) | 90.36 (0.61) |
| Ours + Dropout | **66.97 (0.30)** | 65.14 (0.37) | **64.41 (0.41)** | 87.74 (0.48) | 68.60 (0.62) | 56.80 (0.81) | 78.09 (0.46) | 88.77 (0.65) |
| One-vs-All Baseline | 61.62 (0.31) | 61.30 (0.42) | 58.80 (0.34) | 91.49 (0.37) | 64.09 (1.09) | 51.10 (1.14) | 75.24 (0.84) | 90.81 (0.92) |
| Max. Softmax | 62.43 (0.72) | 62.87 (1.41) | 59.39 (0.61) | 90.92 (0.48) | 65.21 (1.34) | 53.47 (1.45) | 76.01 (1.09) | 89.96 (1.00) |
| Entropy | 63.53 (0.62) | 63.45 (1.36) | 60.59 (0.61) | 90.23 (0.76) | 66.62 (1.51) | 54.37 (1.65) | 77.22 (1.16) | 88.76 (1.16) |
| Bayes-by-Backprop | 64.16 (0.36) | 64.56 (0.61) | 60.64 (0.41) | 90.44 (0.37) | 67.02 (0.84) | 55.60 (1.09) | 76.79 (0.62) | 90.05 (0.57) |
| MC-Dropout | 62.97 (0.22) | 62.21 (0.41) | 60.02 (0.32) | 90.34 (0.37) | 67.19 (1.00) | 56.12 (0.89) | 77.92 (0.92) | 87.38 (1.22) |
| Deep-Ensembles | 66.95 (0.20) | **65.94 (0.39)** | 63.82 (0.10) | 87.90 (0.47) | **71.34 (0.64)** | **59.76 (0.85)** | **80.73 (0.62)** | 84.60 (1.38) |
| Confident Classifier | 62.39 (0.67) | 62.31 (0.24) | 59.78 (0.71) | 90.10 (0.54) | 64.24 (0.83) | 51.61 (0.74) | 75.73 (0.72) | 89.53 (0.94) |
| GEN | 62.66 (0.40) | 61.82 (0.50) | 60.39 (0.48) | 89.89 (0.69) | 62.59 (1.01) | 51.97 (0.51) | 72.87 (1.06) | 93.96 (1.29) |
| Entropy Oracle | 64.42 (0.31) | 65.50 (0.31) | 61.35 (0.37) | 89.87 (0.86) | 70.00 (0.33) | 58.01 (0.24) | 79.95 (0.34) | 85.32 (0.88) |
| One-vs-All Oracle | 67.55 (1.07) | 66.49 (1.24) | 65.17 (1.02) | 86.70 (0.53) | 78.10 (0.99) | 64.04 (1.12) | 86.95 (0.92) | 70.25 (2.55) |
| | CIFAR100 0-49 vs. SVHN | | | | CIFAR100 0-49 vs. Fashion-MNIST | | | |
| Ours | **96.47 (1.26)** | **81.59 (5.55)** | **99.25 (0.26)** | **11.95 (3.93)** | 62.98 (1.71) | 51.89 (2.88) | 73.95 (1.21) | 92.37 (1.04) |
| Ours + Dropout | 95.63 (1.27) | 80.03 (4.04) | 99.17 (0.26) | 15.47 (4.21) | 64.68 (2.02) | 57.46 (1.99) | 74.57 (1.48) | 91.50 (1.15) |
| One-vs-All Baseline | 60.63 (3.90) | 26.37 (6.04) | 87.22 (1.37) | 93.11 (1.92) | 59.80 (2.67) | 51.35 (3.14) | 70.68 (1.29) | 95.13 (0.65) |
| Max. Softmax | 66.32 (2.15) | 37.61 (3.13) | 89.38 (0.78) | 89.73 (1.60) | 68.46 (0.56) | 60.59 (0.85) | 77.66 (0.47) | 89.36 (0.46) |
| Entropy | 68.09 (2.54) | 38.90 (3.51) | 89.84 (0.88) | 89.65 (1.44) | 68.95 (0.48) | 60.59 (0.78) | 78.35 (0.45) | 87.66 (0.75) |
| Bayes-by-Backprop | 72.62 (1.07) | 42.79 (1.62) | 92.21 (0.50) | 81.59 (2.61) | 66.22 (2.02) | 57.55 (3.02) | 75.36 (1.32) | 91.91 (1.20) |
| MC-Dropout | 65.65 (2.51) | 35.78 (3.03) | 88.94 (1.09) | 91.07 (1.95) | **70.50 (1.49)** | **63.75 (1.52)** | 78.70 (1.54) | 88.43 (2.58) |
| Deep-Ensembles | 75.01 (1.06) | 49.77 (2.19) | 91.86 (0.46) | 87.10 (1.26) | 67.42 (1.14) | 60.17 (1.35) | 77.95 (0.83) | **85.46 (1.12)** |
| Confident Classifier | 68.73 (0.49) | 39.41 (0.95) | 90.33 (0.17) | 87.73 (0.71) | 67.04 (0.89) | 56.88 (0.96) | 78.16 (0.38) | 86.27 (1.46) |
| GEN | 92.83 (2.72) | 71.54 (8.20) | 98.58 (0.56) | 23.57 (7.35) | 59.12 (6.18) | 49.44 (5.14) | 70.39 (4.78) | 94.58 (3.06) |
| Entropy Oracle | 86.97 (0.75) | 63.53 (1.22) | 96.97 (0.21) | 50.09 (2.55) | 97.47 (0.45) | 94.63 (0.85) | 98.83 (0.22) | 10.94 (1.96) |
| One-vs-All Oracle | 98.17 (0.22) | 90.82 (0.89) | 99.65 (0.04) | 8.11 (0.98) | 99.62 (0.05) | 99.13 (0.12) | 99.83 (0.02) | 1.65 (0.23) |
| | CIFAR100 0-49 vs. MNIST | | | | | | | |
| Ours | 77.28 (5.81) | 70.28 (6.06) | 84.38 (4.72) | 78.32 (10.93) | | | | |
| Ours + Dropout | 77.13 (3.87) | 73.64 (4.41) | 81.30 (2.83) | 90.07 (2.55) | | | | |
| One-vs-All Baseline | 71.94 (3.24) | 64.45 (3.29) | 78.80 (2.69) | 90.07 (2.67) | | | | |
| Max. Softmax | 75.57 (4.00) | 68.72 (4.65) | 83.09 (2.95) | 81.22 (5.60) | | | | |
| Entropy | 79.14 (4.38) | 72.10 (5.53) | 85.59 (3.17) | 76.56 (8.39) | | | | |
| Bayes-by-Backprop | 71.27 (2.11) | 63.84 (3.33) | 78.08 (1.57) | 91.43 (1.92) | | | | |
| MC-Dropout | 73.41 (1.90) | 66.48 (3.12) | 80.45 (1.38) | 87.39 (2.49) | | | | |
| Deep-Ensembles | **85.92 (1.62)** | **82.02 (1.88)** | **89.77 (1.50)** | **68.03 (5.45)** | | | | |
| Confident Classifier | 77.66 (1.41) | 69.86 (1.33) | 85.11 (0.82) | 76.68 (1.55) | | | | |
| GEN | 77.88 (6.84) | 71.93 (7.67) | 83.30 (5.81) | 80.26 (14.55) | | | | |
| Entropy Oracle | 99.79 (0.06) | 99.56 (0.12) | 99.90 (0.03) | 1.02 (0.28) | | | | |
| One-vs-All Oracle | 99.99 (0.01) | 99.97 (0.01) | 99.99 (0.00) | 0.03 (0.03) | | | | |

Table 11: An OoD-dataset-wise breakdown of the results given in table 5.

| Method | AUROC ↑ | AUPR-In ↑ | AUPR-Out ↑ | FPR@ 95% TPR ↓ | AUROC ↑ | AUPR-In ↑ | AUPR-Out ↑ | FPR@ 95% TPR ↓ |
|---|---|---|---|---|---|---|---|---|
| | Tiny ImageNet 0-99 vs. Tiny ImageNet 100-199 | | | | Tiny ImageNet 0-99 vs. SVHN | | | |
| Ours | 59.16 (0.52) | 61.14 (0.44) | 56.15 (0.40) | 93.10 (0.35) | 98.98 (0.44) | 93.46 (2.87) | 99.81 (0.08) | 3.49 (1.31) |
| Ours + Dropout | **62.01 (0.22)** | **64.70 (0.27)** | **58.16 (0.24)** | **91.97 (0.40)** | **99.39 (0.41)** | **96.18 (2.67)** | **99.89 (0.07)** | **2.19 (1.36)** |
| One-vs-All Baseline | 58.53 (0.45) | 60.28 (0.33) | 55.59 (0.43) | 93.29 (0.27) | 59.65 (4.57) | 32.66 (4.11) | 85.21 (1.86) | 97.02 (0.95) |
| Max. Softmax | 58.16 (0.25) | 60.98 (0.30) | 55.48 (0.28) | 93.22 (0.52) | 63.33 (1.39) | 32.44 (2.05) | 88.51 (0.54) | 90.93 (1.05) |
| Entropy | 58.69 (0.30) | 61.42 (0.31) | 55.68 (0.12) | 93.30 (0.27) | 65.35 (1.48) | 34.25 (1.80) | 88.73 (0.76) | 91.90 (1.66) |
| Bayes-by-Backprop | 57.99 (0.47) | 60.44 (0.42) | 55.25 (0.35) | 93.18 (0.60) | 74.18 (1.78) | 38.32 (1.70) | 93.23 (0.81) | 73.54 (5.25) |
| MC-Dropout | 61.12 (0.40) | 64.16 (0.39) | 57.24 (0.49) | 93.11 (0.62) | 63.54 (4.81) | 37.05 (3.30) | 87.00 (2.21) | 95.33 (1.73) |
| Deep-Ensembles | 60.70 (0.23) | 64.15 (0.23) | 56.84 (0.24) | 93.16 (0.34) | 73.51 (0.81) | 48.67 (1.31) | 90.92 (0.42) | 90.76 (1.18) |
| Confident Classifier | 58.45 (0.23) | 61.26 (0.31) | 55.36 (0.17) | 93.22 (0.53) | 61.86 (1.66) | 31.25 (1.56) | 87.62 (0.61) | 92.79 (0.76) |
| GEN | 56.32 (0.80) | 58.82 (0.48) | 53.80 (0.79) | 94.07 (0.60) | 91.40 (6.55) | 71.24 (16.70) | 98.24 (1.40) | 24.80 (15.25) |
| Entropy Oracle | 58.57 (0.73) | 61.46 (0.75) | 55.49 (0.65) | 93.23 (0.68) | 84.04 (2.72) | 57.38 (4.13) | 96.14 (0.82) | 57.04 (8.27) |
| One-vs-All Oracle | 60.16 (0.31) | 61.23 (0.23) | 57.46 (0.35) | 91.92 (0.57) | 99.39 (0.19) | 95.80 (1.31) | 99.89 (0.03) | 2.25 (0.70) |
| | Tiny ImageNet 0-99 vs. FMNIST | | | | Tiny ImageNet 0-99 vs. MNIST | | | |
| Ours | 55.53 (4.69) | 42.81 (4.04) | 69.36 (3.95) | 93.93 (3.42) | 61.69 (3.34) | 49.40 (3.00) | 73.63 (3.26) | 91.39 (3.62) |
| Ours + Dropout | **95.06 (1.45)** | **89.18 (2.88)** | **97.75 (0.68)** | **17.29 (4.71)** | **99.78 (0.18)** | **99.52 (0.39)** | **99.90 (0.09)** | **1.04 (0.94)** |
| One-vs-All Baseline | 45.63 (4.00) | 41.60 (3.30) | 59.66 (2.04) | 99.70 (0.21) | 51.48 (10.97) | 45.52 (10.95) | 64.34 (7.06) | 97.71 (2.51) |
| Max. Softmax | 61.16 (2.40) | 51.69 (2.62) | 71.73 (1.96) | 94.55 (1.42) | 58.91 (2.05) | 48.66 (2.13) | 70.89 (2.03) | 94.19 (1.70) |
| Entropy | 60.40 (2.88) | 52.07 (2.77) | 69.67 (2.27) | 97.10 (1.16) | 58.77 (2.97) | 49.18 (2.26) | 69.79 (3.04) | 95.64 (2.23) |
| Bayes-by-Backprop | 61.83 (2.86) | 50.29 (2.99) | 73.37 (2.38) | 91.78 (2.26) | 63.37 (5.57) | 48.91 (6.19) | 76.10 (4.20) | 87.10 (5.19) |
| MC-Dropout | 56.34 (6.24) | 53.15 (4.84) | 65.90 (3.83) | 98.79 (0.84) | 71.00 (3.85) | 66.16 (4.66) | 76.05 (2.72) | 95.63 (2.55) |
| Deep-Ensembles | 58.77 (1.34) | 55.05 (2.29) | 66.48 (0.83) | 99.35 (0.25) | 65.32 (1.95) | 59.31 (1.74) | 72.49 (1.96) | 96.40 (1.83) |
| Confident Classifier | 58.59 (2.13) | 49.75 (1.96) | 68.94 (1.61) | 97.04 (0.84) | 57.31 (5.67) | 49.00 (5.73) | 68.20 (3.76) | 96.97 (1.45) |
| GEN | 72.10 (11.72) | 60.00 (14.00) | 83.71 (7.27) | 67.91 (17.06) | 93.78 (4.14) | 87.74 (6.11) | 96.64 (2.87) | 22.91 (18.67) |
| Entropy Oracle | 91.63 (2.01) | 85.61 (3.03) | 95.54 (1.20) | 35.67 (8.17) | 96.29 (1.14) | 93.32 (2.04) | 98.11 (0.61) | 17.98 (5.19) |
| One-vs-All Oracle | 99.81 (0.11) | 99.50 (0.27) | 99.92 (0.05) | 0.74 (0.43) | 100 (0.00) | 100 (0.00) | 100 (0.00) | 0.00 (0.00) |