# OpenReview forum: "UQGAN: A Unified Model for Uncertainty Quantification of Deep Classifiers trained via Conditional GANs"
_NeurIPS.cc/2022/Conference — NeurIPS 2022 Accept_

### Official Review · Reviewer_tm8m · 2022-07-11

**Rating:** 5
**Confidence:** 4
**Soundness:** 3 good
**Presentation:** 2 fair
**Contribution:** 2 fair

**Summary:**

This paper introduces cGANs to help deep classifier modelling uncertainty. The generator is encouraged to generate features that lie in real distribution (evaluated by the discriminator) and out-of-class regions (assessed by the classifier). A low-dimensional regulariser is introduced to generate diverse features. The authors verify their ideas on several benchmarks. The improved performance on out-of-distribution data splits shows the effectiveness of the proposed method. Ablation studies verify the importance of the proposed regulariser.

**Questions:**

-What will happen if the cAE is also trained (not pretrained) during training.

-It seems that the discriminator can easily tell the real features from the fake ones since the distribution of the real features are far away from the fake distribution. In such senario, the generator will occur mode collapse easily as far as my experience. Howevery, mode collapse is not occured in the proposed method. Maybe it's because the low-dimensional regularizer alleviates this issue. Can the authors provide the generated images of the variant in which the low-dimensional regularizer is removed?

**Strengths And Weaknesses:**

Strengths:

-The authors conduct experiments on several datasets, with results showing that the proposed methods are better than previous baselines.

-The authors provide comprehensive ablation studies in the supplementary material to support the soundness of the design of the method.

-The idea of generating out-of-class samples is interesting.


Weakness:

-The visualization experiments are conducted on simple datasets which only contain 2 classes. Authors are encouraged to show some visualization results on datasets having multiple-classes.

-All experiments are conducted on small-scale and small-resolution datasets. The authors are encouraged to verify their method on more challenging datasets, such as ImageNet-Dog, ImageNet-O [1] dataset.

[1] Hendrycks, et al. "Natural adversarial examples." CVPR 2021.

---

> ### Author Response · Authors · 2022-08-02
> **Initial Response**
>
> We included additional toy examples on a 3x3 grid of Gaussians in the appendix of the revised paper version. To give more insight into the performance on higher dimensional data we also included results on TinyImageNet (64x64 images and 200 classes) at the end of the appendix. For the camera ready version the new table 10 will then be moved to the main part of the paper.
>
> Training the conditional autoencoder jointly with the other models was done in GEN, which indicates that this might be viable. However, GANs are usually difficult to train and this can already be observed when the data generating distribution is stationary. Results from statistical learning theory on the convergence of GANs and AEs however support our choice. We also noticed in our experiments on the toy examples that the low dimensional regularizer prevents the mode collapse of the generator. In the appendix you can now also find additional OoC examples with and without the low-dimensional regularizer applied. Especially on the MNIST 0-4 dataset one can notice that the samples are much more diverse when using the regularizer.

---

> > ### Comment · Reviewer_tm8m · 2022-08-09
> > **Response to authors**
> >
> > Thank you for your feedback, which resolves most of my concerns. My rating remains unchanged.

---

### Official Review · Reviewer_pmCh · 2022-07-11

**Rating:** 6
**Confidence:** 2
**Soundness:** 3 good
**Presentation:** 2 fair
**Contribution:** 3 good

**Summary:**

The paper presents UQGAN, a model for uncertainty quantification, including both OoD and FP detection.
The proposed model contains a classifier C that outputs for a pair sample-label $(x,y)$ the probability that the pair is in-distribution.

For training C, the in-distribution pairs consists of real pairs $(x,y)$ directly drawn from the dataset, while out-of-distribution pairs are of two types: either $(x,y')$ where $y'$ is not the correct label associated to $x$, accounting for class uncertainty, or $(\hat{x}, y)$ where $\hat{x}$ is a sample generated by a cGAN conditionned on $y$.
The cGAN's generator is trained in the latent space of a pretrained conditional auto-encoder jointly with a discriminator, but also with the classifier C so that the generated data pair $(\hat{x}, y)$ is considered to be out-of-distribution by C.
Additionally, a regularizer pushes the generated samples to cover a large range of directions in the latent space.

They also combine their method with MC dropout and observe further improvements.

**Questions:**

Looking at the generator objective, it includes a GAN objective, so that the generated distribution p(G(e,y), y) is similar to the real latent distribution p(z,y). This would imply that it pushes $p(\hat{x}, y)$ close to $p(x,y)$, according to a metric induced by the latent space and the discriminator.
The other objective of the generator is given by C, and makes it so that the generated data $(\hat{x}, y)$ is out-of-distribution, essentially pushing away $p(\hat{x}, y)$ and $p(x,y)$.
In some sense, the overall objective of the generator is a compromise between being in-distribution (according to D) and out-of-distribution (according to C).
- Can the authors comment on how much those conflicting objectives are a problem when training the generator? I would guess it would be quite unstable when training for a long time, but that the use of a pre-trained decoder alleviates the issue.
- If we consider the overall objective of the discriminator as a compromise between the two losses, perhaps another way to balance them would be to have the generator target intermediate values for $C(o | \hat{x},y)$ and $D(\hat{x},y)$ such as .75 for instance instead of binary classification. Of course, that would require a different loss for the GAN (and not the wasserstein loss). Have the authors considered something in this direction?



**Limitations:**

The authors have adressed societal impact in the adequate section.

**Strengths And Weaknesses:**

Strength:
- The paper is well organized and didactic. As someone not particularly familiar with the uncertainty literature, I especially appreciate the effort put into the related work section.
- It proposes a novel unified approach for both OoD and FP detection.
- The experiments are comprehensive, with detailed breakdown and hyper-parameter analysis in the supplementary.

Weakness:
- The method seems fairly complex to use as it requires different parts (cAE, Generator/Discriminator, Classifier) and hyper-parameters to tune.

---

> ### Author Response · Authors · 2022-08-02
> **Initial Response**
>
> Our parameter studies in the appendix show that the exact choice of hyperparameters is insignificant, reducing the amount of hyperparameter tuning (please see also our additionally added results on the higher dimensional TinyImageNet dataset which used the same hyperparameters as for CIFAR10/CIFAR100). On the toy examples we observed that the low dimensional regularizer is preventing a mode collapse of the generator. In general, we observed that the training remains stable even when extending the number of epochs. As we observed during our studies, using a Wasserstein formulation for the GAN also helps to stabilize the training. Altering the weight between real and generated OoC examples (coefficient $\lambda_\textrm{real}$) did not affect the stability during training which is why we did not have a look at alternative formulations. We would still like to thank the reviewer for the interesting idea.
>
> Please also have a look at the additional OoC examples, showing qualitatively the influence of the low-dimensional regularizer on the generated samples, as requested by other reviewers. For the camera ready version the new table 10 will be moved to the main body of the paper.

---

> > ### Comment · Reviewer_pmCh · 2022-08-08
> > **Thanks.**
> >
> > Thanks for the response. My questions have been answered, and the additional experiments do reinforce that the setting is likely to be robust. My ratings remain unchanged.

---

### Official Review · Reviewer_PSka · 2022-07-12

**Rating:** 5
**Confidence:** 3
**Soundness:** 2 fair
**Presentation:** 2 fair
**Contribution:** 2 fair

**Summary:**

This paper proposes a method to estimate both aleatoric and epistemic uncertainties. It does so by training a conditional GAN in a latent space of a pretrained autoencoder to shield each class with out-of-class samples.

**Questions:**

Please see my questions in Weakness. Additional question:
- We know that GAN usually produces much higher sharper images than VAEs. However, given the images Fig. 3 are so blurry, and image quality is not the main focus of this paper, can the authors their design choice of using GAN here? Can other generative model be used here?

**Strengths And Weaknesses:**

## Strength
- Evaluation metrics are comprehensive.
- The provided toy dataset is illustrative.

## Weakness
- Some related works are missing, e.g., conformal prediction.
- It is hard to assess the effectiveness of the proposed method without experiments on ImageNet. I understand that the limited computational resources might be a concern, but any form of ImageNet like low resolution of 64x64 or TinyImageNet (which has 200 classes) would be very helpful.
- Section 3.1 and 3.2 are disconnected. How exactly are aleatoric and epistemic uncertainties computed? How are generated $\tilde{x}$ used? Why are they valid definitions?
- Why is one-vs-all classifier necessary here? Can similar results be achieved by normal multiclass classifiers?
- MC-Dropout is basically Bayesian ensemble and can be plugged in other methods as well. It is known that ensemble can drastically improve performance. So the comparisons in Table 3, 4 seem not very fair. Without MC-Dropout, the improvements of the proposed method is not always significant.

---

> ### Author Response · Authors · 2022-08-02
> **Initial Response**
>
> Although conformal prediction methods are conceptually quite remote from our approach, we are ok with extending our related work by citing [a], [b] and [c].
>
> Generated samples $\tilde{x}$ are used in equation (9) as a proxy for out-of-class examples. We compute aleatoric uncertainty as the Shannon entropy over the predicted class probabilites. This is a valid definition because we showed that $\tilde{C}(y|x) \xrightarrow[|S|\to\infty]{} p(y|x)$. If the real $p(y|x)$ is close to a uniform distribution over the classes (which maximizes the Shannon entropy) we have a high uncertainty and vice versa. For epistemic uncertainty $\tilde{C}(i|x)$ can be considered as a proxy for $p(i|x)$, which results in the epistemic uncertainty by $1-\tilde{C}(i|x)$. We thank the reviewer for the remark and will make this clearer by adding the aforementioned explanation in line 161.
>
> For our approach a one-vs-all classifier is necessary to formulate equation (4) ($\tilde{C}(i|x)$). This also allows $\tilde{C}(i|x,y)$ to predict arbitrarily small in-distribution probabilities, which would not be possible with softmax. Additionally this formulation allows us to utilize the out-of-class examples. As these OoC examples can be in-distribution for other classes, this would also not be possible with a multi-class approach.
>
> We apply MC-Dropout to our approach because it is complementing our method in a canonical way by additionally quantifying model uncertainty. If we omit the MC-Dropout we are still improving on the more challenging benchmarks of CIFAR10 0-4 vs. CIFAR10 5-9 and CIFAR100 0-49 vs. CIFAR100 50-99 over the closely related methods "Confident Classifier" and "GEN" especially when considering AUROC, AUPR-In and AUPR-Out.
>
> We choose a GAN for our method as we are relying on the min-max game between the discriminator and classifier to let the generator distribution converge to the boundary of our training distribution. Utilizing other generative models would require changing the respective objective drastically to be able to sample from the boundary. This might even result in additional optimization and preprocessing steps in the pipeline, reducing the end-to-end character of our approach.
>
> In the revised paper version we add additional experiments on the TinyImageNet dataset. You can find the corresponding tables at the end of the appendix. For the camera ready version we will add the new table 10 to the main part of the paper.
>
> [a] Soundouss Messoudi, Sylvain Rousseau, Sébastien Destercke: Deep Conformal Prediction for Robust Models. IPMU 2020
> [b] Maxime Cauchois, Suyash Gupta, John C. Duchi: Knowing what You Know: valid and validated confidence sets in multiclass and multilabel prediction. JMLR (2021)
> [c] Sangdon Park, Osbert Bastani, Nikolai Matni, Insup Lee: PAC Confidence Sets for Deep Neural Networks via Calibrated Prediction. ICLR 2020

---

> > ### Comment · Reviewer_PSka · 2022-08-07
> > **Response to rebuttal**
> >
> > I would like to thank the authors for the rebuttal and providing additional experiment on TinyImageNet. I therefore raise my score from 4 to 5.

---

### Official Review · Reviewer_5piP · 2022-07-18

**Rating:** 6
**Confidence:** 2
**Soundness:** 3 good
**Presentation:** 3 good
**Contribution:** 3 good

**Summary:**

This paper proposes a GAN-based approach to quantify both out-of-distribution and in-distribution uncertainty in image classification. Specifically, the authors use GANs to generate examples from the out-of-class regime several times, and each time then use the examples to train a one-vs-all classifier in the final DNN layer. Finally, the resulting classifiers are sued to model class conditional likehoods. The authors conduct experiments on MNIST, CIFAR10， CIFAR100 and show state-of-the-art performance in terms of OoD detection and FP detection.

**Questions:**

1. Typically, GANs are trained to model the distribution of real data, and sometimes even humans cannot distinguish real data from examples generated by state-of-the-art GANs [1].  My question is how to determine whether a GAN-generated example is out-of-distribution or in-distribution? How to choose the GAN models? How does the quality of GAN-generated examples affect uncertainty quantification?

[1] Karras T, Laine S, Aittala M, et al. Analyzing and improving the image quality of stylegan[C]//Proceedings of the IEEE/CVF conference on computer vision and pattern recognition. 2020: 8110-8119.

2. Although the authors provide a survey [16] on uncertainty quantification methods in Line 28, we suggest a brief introduction to uncertainty quantification in the section of Related Work. Besides, we suggest breaking the section of Related Work into several subsections, which can make it clear.


**Ethics Review Area:**

["I don’t know"]

**Limitations:**

1. The organization of the section of Related Work is not clear.
2. It would be better to show more GAN-generated examples for reference. Besides, it would be better to conduct experiments on a relatively large dataset, e.g., ImageNet.


**Strengths And Weaknesses:**

Strengths：
1. The presentation of this paper is clear and easy to follow.
2. Existing GAN-based approaches mostly only predict a single score for OoD examples, while this paper proposes to quantify both in-distribution uncertainty and out-of-distribution uncertainty. This paper achieves this by repurposing the training scheme of the classifier and the generation of OoD examples. The idea is interesting and can inspire others in this community.
3. The authors show strong results in several datasets, including MNIST, CIFAR10， CIFAR100.

Weaknesses:
1. There are no ablation studies to study and verify the effectiveness of each design of the proposal. For example, the number of classes used for training the classifier each time, the different image quality of GAN-generated examples, etc.
2. It would be better to clarify the necessity and importance (or practical significance) of quantifying both in-distribution and out-of-distribution uncertainty in a single model.

---

> ### Author Response · Authors · 2022-08-02
> **Initial Response**
>
> In terms of the influence of quality of generated OOD examples, we tested generator networks with different numbers of parameters, which should also alter the quality of generated samples, and noticed low to no influence on the OoD detection performance of the final classifier. We will address this with a short note in the appendix.
> By design of the min-max game between the discriminator and the classifier in the GAN objective, the generator produces samples on the boundary of the training distribution which can then be utilized as OoC examples. To some extent the loss weight for the classifier, $\lambda_\textrm{cl}$, is also controlling the closeness to the training distribution. To provide some insight into the influence of the number of classes, we will add an ablation study showing the performance of the UQGAN classifier in comparison to the entropy baseline dependent on the number of classes.
>
> Our related work is organized such that we reference common methodology and publications that are related to the different parts of our method in dedicated paragraphs. For the camera ready version we will add paragraph headings to make this structure clearer and add a small introduction as follows:
>
> "In this section we give an overview of common uncertainty quantification methods as well as publications related to the different parts of our method. In this context the task of uncertainty quantification is to assign scalar values to predictions, quantifying aleatoric (in-distribution) and epistemic (out-of-distribution) uncertainty."
>
> We thank the reviewer for the constructive remark.
>
> Please see the appendix for more generated OoC examples with and without the low-dimensional regularizer. Additionally we supplied results on the TinyImageNet dataset (64x64 images and 200 classes) which is well beyond the evaluation of the cited publications. For the camera ready version the new table 10 will then be moved to the main body of the paper.

---

> > ### Comment · Reviewer_5piP · 2022-08-08
> > **Thanks for the authors' response**
> >
> > Thanks for the response.
> > Most of my concerns have been addressed and I will keep my rating.
> > We suggest the authors polish the final version according to the response if it is accepted.

---

### Meta-Review · Area_Chair_L4tS · 2022-08-30

**Recommendation:** Accept
**Confidence:** Less certain

**Metareview:**

The authors propose a new approach for training image classifiers with complete uncertainty quantification based on generative adversarial networks. The main idea is to use GANs to "shield" each class separately from the out-of-class (OoC) regime. This is done in combination with a one-vs-all classifier in the final DNN layer trained jointly with a class-conditional generator for out-of-class data in an adversarial framework. Finally, these classifiers are then used to model class conditional likelihoods. The empirical validation shows improved OoD detection and FP detection performance when compared to SOTA in this setting.

The reviewers appreciated the clarity of exposition and the positioning with respect to the related works. The unified approach applicable both to FP detection and OoD detection was deemed novel. On the negative side, the method seems to be extremely involved in terms of the required architectural pieces, distinction between low-dim and high-dim settings, primarily low-resolution data used for evaluation, and the number of hyperparameters. During the discussion the authors addressed the main questions raised by the reviewers. Nevertheless, given that all of the reviewers are leaning positive, I'll recommend the acceptance of this work. Please do a full pass in terms of formatting of the whole manuscript, including removing inline tables and figures, removing things like double parenthesis, bolding specific letters (e.g. L247), clarify the flow of information in figure 1 so that one can grasp the high-level overview of the algorithm, and incorporate the remaining points raised during the discussion.

**Award:**

No

---

### Decision · Program_Chairs · 2022-09-14

Accept